EMBO
Molecular Medicine

# Zinc inhibits lethal inflammatory shock by preventing microbe-induced interferon signature in intestinal epithelium

Jolien Souffriau[1,2,†], Steven Timmermans[1,2,†], Tineke Vanderhaeghen[1,2], Charlotte Wallaeys[1,2], Kelly Van Looveren[1,2], Lindsy Aelbrecht[1,2], Sylviane Dewaele[1,2], Jolien Vandewalle[1,2], Evy Goossens[3], Serge Verbanck[3], Filip Boyen[3], Melanie Eggermont[1,2], Lindsey De Commer[4,5], Riet De Rycke[6,7], Michiel De Bruyne[6,7], Raul Tito[4,5], Marlies Ballegeer[1,2], Sofie Vandevyver[1,2], Tiago Velho[8], Luis Ferreira Moita[8], Tino Hochepied[1,2], Karolien De Bosscher[9,10], Jeroen Raes[4,5], Filip Van Immerseel[3], Rudi Beyaert[1,2] & Claude Libert[1,2,*]

## Abstract

The cytokine TNF drives inflammatory diseases, e.g., Crohn's disease. In a mouse model of TNF-induced systemic inflammatory response syndrome (SIRS), severe impact on intestinal epithelial cells (IECs) is observed. Zinc confers complete protection in this model. We found that zinc no longer protects in animals which lack glucocorticoids (GCs), or express mutant versions of their receptor GR in IECs, nor in mice which lack gut microbiota. RNA-seq studies in IECs showed that zinc caused reduction in expression of constitutive (STAT1-induced) interferon-stimulated response (ISRE) genes and interferon regulatory factor (IRF) genes. Since some of these genes are involved in TNF-induced cell death in intestinal crypt Paneth cells, and since zinc has direct effects on the composition of the gut microbiota (such as several *Staphylococcus* species) and on TNF-induced Paneth cell death, we postulate a new zinc-related anti-inflammatory mechanism. Zinc modulates the gut microbiota, causing less induction of ISRE/IRF genes in crypt cells, less TNF-induced necroptosis in Paneth cells, and less fatal evasion of gut bacteria into the system.

**Keywords** genetics; inflammation; microbiota; nutrient; regulation

**Subject Categories** Digestive System; Immunology; Microbiology, Virology & Host Pathogen Interaction

## Introduction

The cytokine TNF is a central player in inflammatory diseases and infections. TNFR1 (Puimege *et al*, 2014) is the main pathogenic TNF receptor and is strongly stimulated by acute and chronic expressed TNF. The systemic effects of TNF have been studied in detail. It is striking that the intestinal epithelium is sensitive to TNF-induced damage: Transgenic TNF overexpression, via a myriad of mechanisms, clearly leads to inflammatory bowel disease and/or arthritis, and injection of TNF in mammals has shown that intestinal toxicity is a major dose-limiting issue (Piguet *et al*, 1998; Kontoyiannis *et al*, 1999). Cell death of IECs as well as reduced expression of tight junctions appears essential in the increase in gut permeability induced by TNF. This reduced barrier between the homeostatic body and the gut luminal content is crucial (Van Hauwermeiren *et al*, 2015). In the lumen, the microbiota, consisting of many billions of microbes,

1  Center for Inflammation Research, VIB, Ghent, Belgium
2  Department of Biomedical Molecular Biology, Ghent University, Ghent, Belgium
3  Department of Pathology, Bacteriology and Avian Diseases, Faculty of Veterinary Medicine, Ghent University, Merelbeke, Belgium
4  Department of Microbiology and Immunology, Rega Institute, KU Leuven, Leuven, Belgium
5  VIB-KU Leuven Center for Microbiology, Leuven, Belgium
6  Department of Biomedical Molecular Biology and Expertise Centre for Transmission Electron Microscopy, Ghent University, Ghent, Belgium
7  VIB Center for Inflammation Research and BioImaging Core, VIB, Ghent, Belgium
8  Instituto Gulbenkian de Ciência, Oeiras, Portugal
9  VIB Center for Medical Biotechnology, Ghent, Belgium
10  Department of Biochemistry, Ghent University, Ghent, Belgium
   *Corresponding author. Tel: +32 9 3313700; E-mail: claude.libert@irc.vib-ugent.be
   †These authors contributed equally to this work as first authors
   The authors want to dedicate this paper to the memory of Prof. Walter Fiers (1931–2019), who passed away on 31 July 2019, and who was mentor and strong supporter of this project and research team.

predominantly bacteria, is present (Vrancken *et al*, 2019). Most of these are harmless if they remain in the gut lumen. TNF has been shown to lead to outflow of bacteria from the lumen and colonization of draining mesenteric lymph nodes (MLNs) and spleen (Van Hauwermeiren *et al*, 2015). Hence, broad-spectrum antibiotics confer significant protection against TNF-induced lethal systemic inflammatory response syndrome (SIRS; Van Hauwermeiren *et al*, 2015). Since IEC-specific depletion of TNFR1 leads to a similar protective effect as an antibiotic treatment (Van Hauwermeiren *et al*, 2013), it appears conceivable that TNF, via TNFR1, induces IEC damage, followed by a bacterial contamination of organs.

The damage that is caused by TNF to IECs is incompletely understood, but is of utmost importance (Van Assche *et al*, 2010), as it may play a role in Crohn's disease. TNF induces cell death of IECs, shrinkage of villi and erosion (Piguet *et al*, 1998), and death of goblet cells as well as Paneth cells (Van Hauwermeiren *et al*, 2015). Glucocorticoids (GCs; e.g., the synthetic dexamethasone (Dex)), which function by binding to the glucocorticoid receptor (GR), confer protection against Crohn's diseases (Van Assche *et al*, 2010) and against TNF-induced intestinal damage and lethal shock, when given prior to TNF. From a therapeutic viewpoint, this preventive therapy is not relevant, but this effect can help elucidating essential mechanisms. It was also shown that removal of adrenals which produce endogenous GCs drastically sensitizes to TNF and that GR mutant mice have sensitized IECs toward TNF (Ballegeer *et al*, 2018). The mechanism by which GCs/GR dampens TNF effects in the gut is thought to relate to the gut commensal flora: The microbes were recently shown to chronically induce, in the ileum, interferon-stimulated response element (ISRE) genes, and interferon regulatory factor (IRF) genes, some of which are involved in necroptosis (*Ripk3*, *Mlkl*, and *Zbp1*; Ballegeer *et al*, 2018). When GCs or GR was absent, the IECs lost control over these genes, and hence, these genes underwent high expression. Under such conditions, low doses of TNF sufficed to cause induction of necroptosis followed by death of the animals. Interestingly, this necroptosis was confined to crypt cells known as Paneth cells (Ballegeer *et al*, 2018).

In humans, 25% of the global mortalities are caused by microbial infections. Many of these problems are due to the development of resistance against antibiotics. Investigation of interesting alternative antimicrobial strategies has become mandatory. Zinc has a huge potential in this respect, since zinc has been shown to increase the resistance of vulnerable patients (children, elderly, underfed) against diarrhea and gastrointestinal infections (Souffriau & Libert, 2018). We have previously published that a pretreatment of mice with $ZnSO_4$ in the drinking water for 1 week leads to strong protection against TNF-induced SIRS (Waelput *et al*, 2001; Van Molle *et al*, 2007), but the mechanism of protection remained unclear. Zinc is essential for a healthy intestinal homeostasis and is known to be essential for Paneth cell function and survival (Jouppila *et al*, 1976; Podany *et al*, 2016). In these cells, zinc is actively adsorbed by transporters and shuttled into the secretory vesicles, which contain anti-bacterial defensins, some of which are activated by zinc-dependent proteases, the most important of which being matrix metalloproteinase-7 (MMP7; Wilson *et al*, 1999). Zinc has been shown to activate a specific transcription factor, metal transcription factor 1 (MTF1) by direct binding (Gunther *et al*, 2012), leading to numerous transcriptional changes, e.g., induction of a zinc

transporter known as ZnT2 and coded by *Slc30a2*. ZnT2 is of importance in the intracellular zinc transport in Paneth cells (Podany *et al*, 2016). Zn has also been shown to have profound effects on the composition of the gut microbiota (Li *et al*, 2016; Zackular *et al*, 2016), but whether this is via direct anti-bacterial effects, or via Paneth cells or other mechanisms is not clear.

In this paper, we studied the mechanism of protection of zinc against TNF-induced lethal SIRS. We studied the cross-talk of zinc with GCs/GR and found that zinc is unable to protect against TNF when GR is mutated or absent in the IECs or when corticosterone is absent. Zinc appeared to reduce the ISRE/IRF-dependent genes in the IECs, thereby protecting these cells (particularly Paneth cells) against TNF-induced necroptosis and bacterial influx from the gut lumen into the system. The mechanism by which zinc reduces ISRE/IRF genes was investigated and appears to relate to direct anti-bacterial effects of zinc against certain bacterial taxa, such as *Staphylococcus sciuri* and *Staphylococcus nepalensis*. The elucidation of the action mechanism of zinc may lead to better understanding of zinc's effects in the treatment of intestinal infections and diarrhea in humans and farm animals.

## Results

### Glucocorticoids and GR dimers play an essential role in the zinc protection against TNF-induced lethality

Endogenous and synthetic GCs and their receptor GR are crucial for the survival of mice against TNF-induced SIRS (Vandevyver *et al*, 2012; Ballegeer *et al*, 2018). We investigated whether GCs are involved in the zinc-induced protection against TNF. We used adrenalectomized (Adx) mice, which are unable to produce corticosterone. Adx mice were pretreated for 7 days with 25 mM $ZnSO_4$ via the drinking water, a protocol that was published to confer optimal protection in the TNF model in normal mice (Waelput *et al*, 2001). In contrast to naïve C57BL/6J mice, Adx mice were not protected by $ZnSO_4$ against a lethal intraperitoneal (i.p.) injection of TNF (Fig 1A). Serum corticosterone levels were indeed very low in Adx mice and were not increased upon 1-week treatment of mice with $ZnSO_4$ (Fig 1B). $GR^{Dim}$ mice, which express a point-mutant GR protein that forms less efficient GR homodimers and DNA interaction, were previously found to be extremely sensitive for TNF-induced SIRS (Reichardt *et al*, 1998; Van Bogaert *et al*, 2011; Vandevyver *et al*, 2012; Ballegeer *et al*, 2018). $ZnSO_4$ was unable to confer protection against TNF in these mice (Fig 1C). Increasing the zinc dose from 25 to 75 mM (the maximal tolerated dose for mice) for a week had no protective effect in $GR^{Dim}$ mice either. Therefore, we conclude that the zinc protection against TNF requires the presence of corticosterone and functional GR dimers, i.e., involves GR-mediated gene regulation.

We further confirmed that the complete lack of protection of zinc in Adx and $GR^{Dim}$ mice is not reflected in a lack of zinc uptake by these mice, since blood zinc levels 1 week after 25 mM $ZnSO_4$ were equally increased in these as in control mice (Fig 1D and E). Blood zinc levels fluctuate in mammals between 75 and 125 µg/dl (Reyes, 1996). In our studies, 1 week of treatment increases these levels to 200–300 µg/dl. Zinc is able to upregulate the mRNA expression of several genes (see further Fig 2) in the ileum of mice, and several

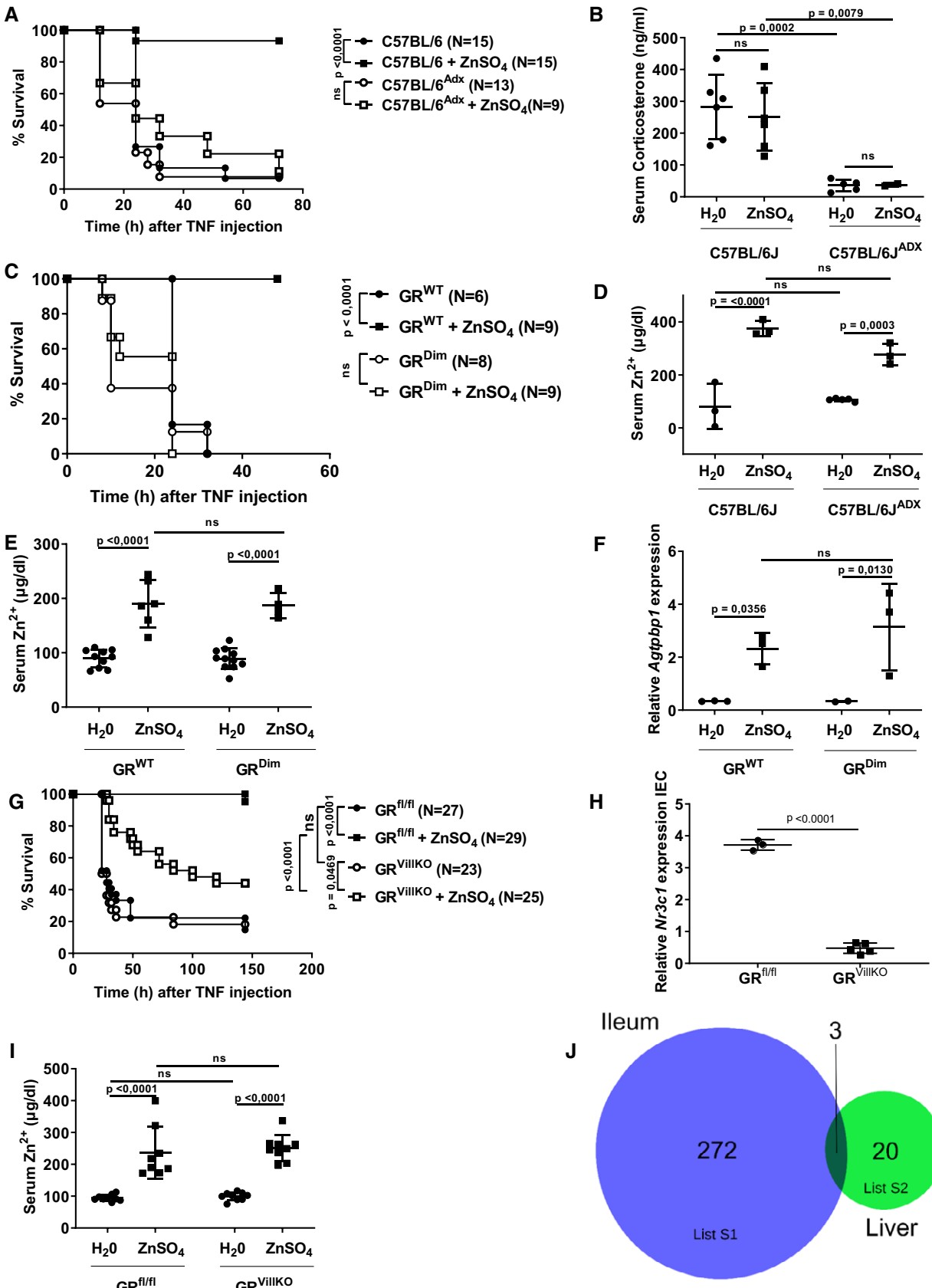

**Figure 1.**

**Figure 1.  ZnSO₄ fails to protect against a lethal TNF injection in Adx, GR^Dim, and GR^VilIKO mice.**

A   Survival curves of C57BL/6J naïve and adrenalectomized (Adx) mice treated for 7 days with 25 mM ZnSO₄ in the drinking water and challenged, i.p., with 30 μg (naïve) or 2 μg (Adx) TNF, solved in sterile PBS, per 20 g bodyweight. No deaths occurred later than 72 h after TNF injection. Results represent combined data of three experiments.

B   Corticosterone levels in the serum of C57BL/6J naïve and Adx mice after 7 days of 25 mM ZnSO₄ supplementation to the drinking water (N = 6 per group).

C   Survival curves of GR^WT and GR^Dim mice treated for 7 days with 25 mM ZnSO₄ in the drinking water and challenged, i.p., with 50 μg (GR^WT) or 12.5 μg (GR^Dim) TNF, solved in sterile PBS, per 20 g bodyweight. No deaths occurred later than 50 h after TNF injection. Results represent combined data of two experiments.

D, E   Zinc levels in the serum after 7 days of 25 mM ZnSO₄ supplementation to the drinking water of C57BL/6J naïve and Adx mice (D, N = 3–6), or GR^WT and GR^Dim mice (E, N = 6–9).

F   *Agtpbp1* gene expression measured with RT–qPCR in the ileum of GR^WT and GR^Dim mice treated for 7 days with 25 mM ZnSO₄ in the drinking water (N = 3 per group).

G   Survival curves of GR^fl/fl and GR^VilIKO mice treated for 7 days with 25 mM ZnSO₄ in the drinking water and challenged, i.p., with 35 μg TNF, solved in sterile PBS, per 20 g bodyweight. No deaths occurred later than 150 h after TNF injection. Results represent combined data of six experiments.

H   Mouse *Nr3c1* gene expression measured with RT–qPCR in the intestinal epithelial cells (IECs) of GR^fl/fl and GR^VilIKO mice (N = 3–6). *Hprt* and *Villin* were used as housekeeping genes.

I   Zinc levels in the serum after 7 days of 25 mM ZnSO₄ supplementation to the drinking water of GR^fl/fl and GR^VilIKO mice (N = 9 per group).

J   Number of genes differentially expressed (up and down) by zinc (LFC > 0.8 and P < 0.05) measured by RNA sequencing in the ileum and liver of C57BL/6J mice treated with 25 mM ZnSO₄ in the drinking water for 7 days (N = 3).

Data information: For the survival curves, *P*-values were analyzed with a chi-square test. In (B, D, E, F, H, I), the data are shown as mean ± SD. *P*-values were analyzed with Student's *t*-test (unpaired, two-tailed) on the log-transformed data in (B, H) and with a two-way ANOVA followed by a Tukey multiple comparisons test in (D, E, F, I). Significant expression of genes in the RNA sequencing was assessed with a Wald test with negative binomial distribution in DESEQ2.

key control genes, e.g., *Agtpbp1*, were found to be perfectly induced by zinc in GR^Dim as in GR^WT mice, illustrating that zinc in these GR^Dim mice does have biological activities (Fig 1F).

## Zinc fails to efficiently protect against TNF when intestinal epithelial GR is deleted

We recently reported that the expression of GR in the IECs is essential in conferring resistance against TNF-induced SIRS (Ballegeer *et al*, 2018). We applied GR^VilIKO mice and observed a clearly less robust protection by zinc as compared to GR^fl/fl mice (Fig 1G). The residual protection of zinc in the GR^VilIKO mice can result from suboptimal penetrance of the villin promoter (Fig 1H), although protective effects of zinc in other cell types cannot be excluded as an explanation. Zinc uptake in these mice was confirmed (Fig 1I). To further study the impact of zinc on the intestinal epithelium, bulk RNA-seq was performed on liver samples and on ileum samples of C57BL/6J mice, treated for a week with 25 mM ZnSO₄ or normal water. Throughout this paper, the group sizes of RNA-seq experiments were *n* = 3, which is a minimum, but the quality of sequences was superb. Zinc caused differential regulation of 275 genes in the ileum and 23 genes in the liver (Fig 1J and Appendix Tables S1 and S2), with only three genes overlapping (*Mt1*, *Mt2*, and *Slc39a4*), which is compatible with a dominant role of ileum as a main zinc-responsive organ.

## Zinc does not modulate GR transcriptional activity in a direct way

The GR protein contains 2 zinc-finger motives in the DNA-binding domain, and both are essential for GR DNA-binding, GR homo-dimerization, and GR-mediated gene regulation (Vandevyver *et al*, 2014). We studied if zinc administration via supplementation of the GR zinc fingers optimizes the GR dimer-mediated gene transcription. The impact of zinc on the transcription of GRE genes was studied on the genome-wide level via RNA-seq in the ileum of C57BL/6J mice. Mice were treated with normal drinking water or 25 mM ZnSO₄ drinking water (1 week) followed by a single intraperitoneal

(i.p.) injection of Dex (200 μg) or PBS. A 1-week zinc pretreatment or 2-h Dex pretreatment is optimal times to protect against TNF. We found no convincing evidence for a direct GR-(co)activation role of zinc: Fig 2A illustrates the overlap between zinc-upregulated and Dex-upregulated genes in the ileum. Only 4.2% of the genes upregulated by Dex could also be induced by zinc (20/481 genes). Also, at the level of downregulated genes, only a small overlap (4.98%) was noticed between genes downregulated by zinc or by Dex (Fig 2B). Because of the small overlap, we believe it is unlikely that zinc (co-) activates GR transcription as a ligand, agonist, cofactor, or zinc-finger stabilizer.

We further investigated if zinc has an impact on the induction levels of Dex-modulated (GRE) genes. The schemes in Fig 2C (related to Dex-induced genes) and Appendix Fig S1 (related to Dex-reduced genes) demonstrate that of all the genes that are exclusively regulated by Dex, only a small percentage is further induced or reduced by the addition of zinc. Since these genes form only 3.8% (17/451) of the Dex-induced genes and only 6.5% of the Dex-reduced genes (16/247), we believe that zinc does not increase general GR activity. The identities of these genes and their counts and log-fold changes are provided in Appendix Tables S3–S16. In Appendix Fig S2, we provide two examples of each informative category of the Dex-induced GRE genes, non-Dex-induced genes, and the impact of ZnSO₄ on their expression levels according to the 8 classes that we defined in Fig 2C, by RNA-seq. There are a number of gene families of interest, for example, list S9, which are genes induced exclusively by the combination of Dex and ZnSO₄.

We conclude that zinc protects against TNF-induced lethality in a way that requires the presence of GCs and GR (dimers), in the ileum, but not through a general increase in GR-induced transcription. To make this statement harder, we performed RNA-seq experiments to study the Zn-induced and Zn-repressed signatures in GR^WT and in GR^Dim mice. We have chosen these mutants above GR^VilIKO because of the observed imperfect cre penetrance with villin-cre. We have compared the induction of the 116 genes of Fig 2A in GR^WT and GR^Dim mice, and we found that the majority of these genes (*n* = 67) is also significantly induced in GR^Dim mice, while the other 49 genes are also induced but do not reach significance. By

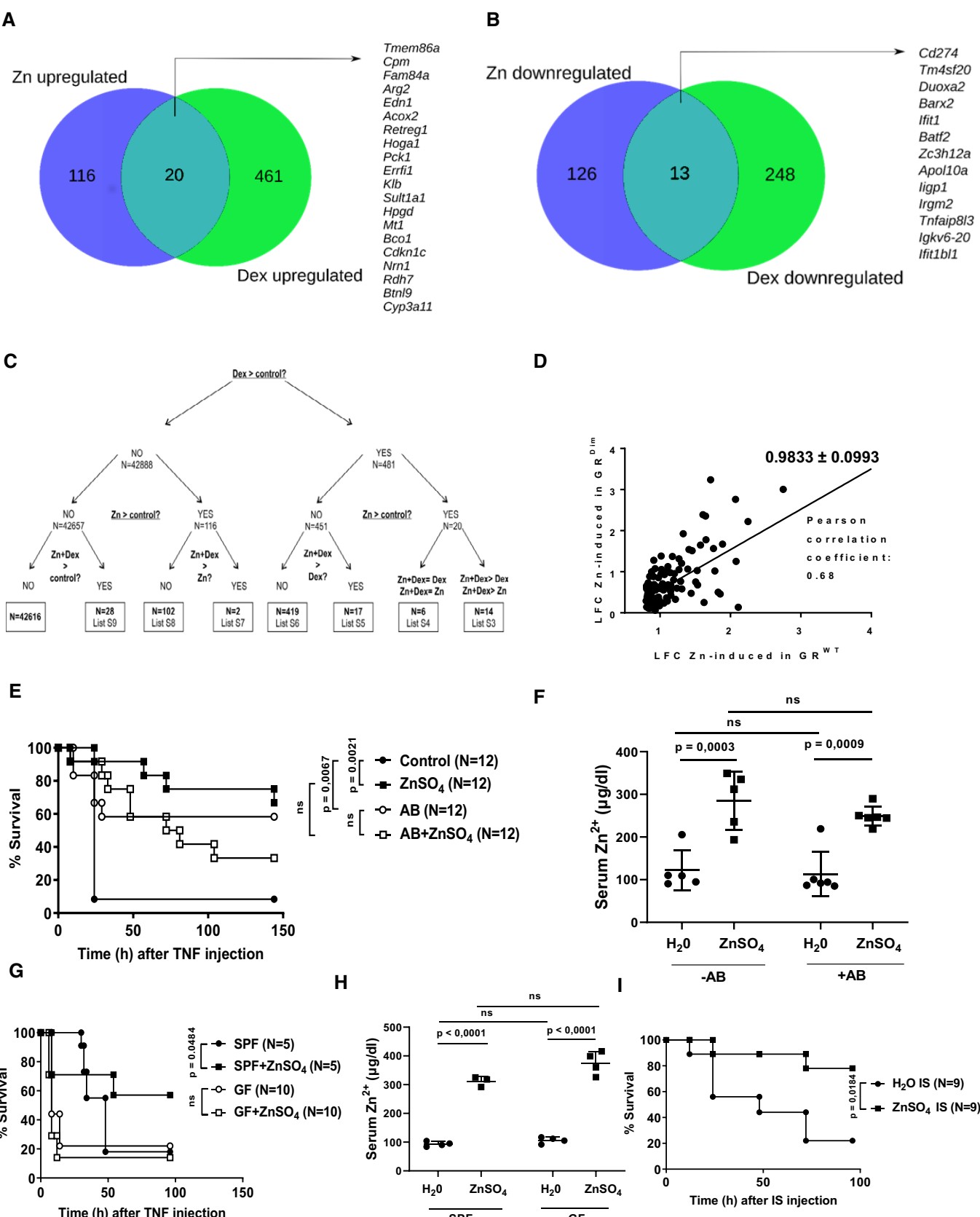

**Figure 2.**

**Figure 2. Transcriptional effects of ZnSO₄ in the ileum and the role of antibiotics and germ-free aspects.**

A–C  RNA sequencing on ileum samples of C57BL/6J mice treated with water or 25 mM ZnSO$_4$ water for 7 days and then injected i.p. with 200 µg Dex (Rapidexon) solved in 200 µl PBS, or with PBS only. 2 h after injection, the ileum was isolated ($N$ = 3 per group). Overlap between genes (A) upregulated or (B) downregulated by ZnSO$_4$ (Zn) or Dex (LFC > 0.8, $P$ < 0.05). (C) Scheme of numbers of genes upregulated in ZnSO$_4$, Dex, or ZnSO$_4$+Dex-treated mice (LFC > 0.8, $P$ < 0.05). Significant expression of genes in the RNA sequencing was assessed with a Wald test with negative binomial distribution in DESEQ2.

D  Regression curve plotting LFCs of genes induced by ZnSO$_4$ in GR$^{WT}$ and GR$^{Dim}$ mice, as measured by RNA-seq in ileum samples. All genes, induced in ileum by DEX in GR$^{WT}$ with LFC > 0.8, $P$ < 0.05 are considered, while these genes with $P$ < 0.05 in GR$^{Dim}$ were considered. The slope of the correlation curve as well as Pearson correlation coefficient was calculated by GraphPad Prism.

E  C57BL/6J mice received antibiotics (AB) in the drinking water for 3 weeks, followed by 1 week of 25 mM ZnSO$_4$ in the drinking water. During this week, AB administration was continued by oral gavage. Mice were challenged i.v. with 12.5 µg TNF solved in sterile PBS and survival recorded. Combined data of two experiments.

F  Zinc levels measured in serum after the antibiotics and ZnSO$_4$ protocol ($N$ = 5–8).

G,H  Mice housed in germ-free (GF) or specific pathogen-free (SPF) conditions received 25 mM ZnSO$_4$ in the drinking water for 1 week and were then challenged i.p. with 35 µg TNF, dissolved in sterile PBS, per 20 g bodyweight. Combined data of two experiments. (H) displays the blood zinc levels in SPF and GF mice treated for 1 week with water or ZnSO$_4$ ($N$ = 3/4 per group).

I  Ileum Slurry (IS) was isolated from C57BL/6J mice put on 25 mM ZnSO$_4$ or control water for 1 week, and 750 µl of this IS was injected i.p. in C57BL/6J GF mice.

Data information: For the survival curves, $P$-values were analyzed with a chi-square test. In (F) and (H), data are shown as mean ± SD and $P$-values were analyzed with a two-way ANOVA.

chi-square analysis with Yates correction, or Fisher exact test, this high degree of resemblance of Zn-induction patter in GR$^{WT}$ and GR$^{Dim}$ has $P$ < 0.0001. Also, plotting the Zn-induction gene levels observed by RNA-seq in GR$^{Dim}$ and GR$^{WT}$ and detection of the regression curve (Fig 2D), we observe a slope of 0.9833, suggesting a near-perfect correlation of both datasets, and a Pearson correlation coefficient of 0.68, which is considered as a strong correlation. These data suggest that indeed Zn-upregulated genes are equally strong induced whether GR is active or not. Very similar data are obtained in case of Zn-repressed genes.

## Impact of zinc on inflammation and microbial virulence

Zinc has been shown to exert anti-inflammatory effects (Souffriau & Libert, 2018). Since the TNF-induced lethal SIRS model is primarily inflammatory, we studied whether the protective effect of zinc pretreatment is reflected in anti-inflammatory readouts, which may couple zinc to the anti-inflammatory GC/GR axis. The expression of 12 key cytokines, systemic and local, including IL1β, as well as soluble TNF receptors revealed that zinc protection was not related to an obvious reduction in systemic or IEC-local inflammatory profiles. The induction by TNF of the serum levels of two cytokines were reduced by zinc, i.e., IL-6 and Eotaxin (Appendix Fig S3). Older work from our group, using IL6$^{-/-}$ and IL6 receptor$^{-/-}$ mice, has revealed that this cytokine is not involved in the TNF model as a mediator (Libert *et al*, 1992), while the limited effect of zinc on Eotaxin is unlikely to contribute to the robust protection zinc exerts (Libert *et al*, 1994).

Mice pretreated with an antibiotics cocktail in the drinking water to deplete the intestinal microbiome are protected against TNF (Van Hauwermeiren *et al*, 2013, 2015), because it is thought that bacteria contribute to TNF-induced lethality because they penetrate through the IEC breaches (generated by TNF) and colonize organs as spleen and mesenteric lymph nodes. ZnSO$_4$, which protected C57BL/6J mice against TNF, was unable to confer further protection in antibiotic-treated mice (Fig 2E), while zinc levels were equally increased in the blood of both ZnSO$_4$-treated groups of mice (Fig 2F). Interestingly, ZnSO$_4$ was also unable to protect against TNF in C57BL/6J mice, born, and raised in germ-free (GF) conditions (Fig 2G), despite 1 week of ZnSO$_4$ treatment led to equal increases in blood

zinc levels in GF mice (Fig 2H). The failure of zinc to protect was not related to failure of mice to drink or to dehydration. These data suggest that zinc modulates the microbiome, because in the absence of gut flora, the target of zinc is no longer present, and zinc has become irrelevant. We must remark that antibiotic-treated mice are significantly protected against TNF, as published earlier (Dejager *et al*, 2015), but that GF mice do not show such a protection. This was confirmed by TNF titration studies in normal bred C57BL/6J and GF C57BL/6J.

As a first step to study if zinc treatment of mice leads to a reduction in virulence of the gut microbiota, we isolated ileum contents (ileum slurry, IS) from C57BL/6J mice, treated with normal drinking water or with 25 mM ZnSO$_4$ during a week. Ileum slurry was prepared and normalized on an exact identical w/v basis from both donor populations (as described by (Starr *et al*, 2014)) and was injected in naïve germ-free (GF) C57BL/6J receptor mice, 750 µl/mouse and lethality recorded. Mice receiving the IS isolated from ZnSO$_4$-treated mice showed significantly less lethality than mice receiving the IS isolated from control mice (Fig 2I).

## Investigation of transcriptional hubs in IECs involved in zinc-mediated protection against TNF

Since the zinc protection is outspokenly depending on the transcription factor activity of GR, we first set out to investigate transcriptional modules in IECs that might be linked to (anti)microbial effects. Paneth cells are known as the master regulators, mainly by the release of antimicrobial peptides (AMPs) in the crypts of the ileum (Wilson *et al*, 1999; Ouellette, 2011). These cells contain secretory granules in which AMPs are stored and matured. Paneth cells contain zinc and depend on zinc for survival. It was also shown that zinc collaborates, via MTF1, with GCs (via GR) for the induction of the *Slc30a2* gene, which encodes the zinc transporter ZnT2 (Guo *et al*, 2010) which imports zinc from the cytoplasm into the secretory granules. There, zinc is thought to help in the maturation or stabilization of AMPs (Podany *et al*, 2016), possibly by activating zinc-dependent matrix metalloproteinase 7 (MMP7) (Wilson *et al*, 1999). We studied the protection of zinc in MTF1$^{VillKO}$ mice, in Slc30a2$^{-/-}$ and in MMP7$^{-/-}$ mice but found no evidence for a role of this axis in the protection of zinc against TNF (Appendix Fig S4).

**A**

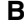

| Rank | Motif | GR<sup>WT</sup> | Name | P-value | log P-pvalue |
|---|---|---|---|---|---|
| 1 | | AGTTTCAGTTTC | ISRE(IRF)/ThioMac-LPS-Expression(GSE23622)/Homer | 1e-11 | -2.562e+01 |
| 2 | | GAAAGTGAAAGT | IRF1(IRF)/PBMC-IRF1-ChIP-Seq(GSE43036)/Homer | 1e-8 | -1.931e+01 |
| 3 | | AGTTTCAGTTTC | IRF3(IRF)/BMDM-Irf3-ChIP-Seq(GSE67343)/Homer | 1e-6 | -1.382e+01 |
| 4 | | GAAAGTGAAAGC | IRF2(IRF)/Erythroblas-IRF2-ChIP-Seq(GSE36985)/Homer | 1e-5 | -1.337e+01 |
| 5 | | GGAAGTGAAAGT | PU.1:IRF8(ETS:IRF)/pDC-Irf8-ChIP-Seq(GSE66899)/Homer | 1e-4 | -1.114e+01 |

| Rank | Motif | GR<sup>Dim</sup> | Name | P-value | log P-pvalue |
|---|---|---|---|---|---|
| 1 | | AGTTTCAGTTTC | ISRE(IRF)/ThioMac-LPS-Expression(GSE23622)/Homer | 1e-14 | -3.301e+01 |
| 2 | | GAAAGTGAAAGT | IRF2(IRF)/Erythroblas-IRF2-ChIP-Seq(GSE36985)/Homer | 1e-8 | -2.061e+01 |
| 3 | | GAAAGTGAAAGT | IRF1(IRF)/PBMC-IRF1-ChIP-Seq(GSE43036)/Homer | 1e-8 | -2.024e+01 |
| 4 | | AGTTTCAGTTTC | IRF3(IRF)/BMDM-Irf3-ChIP-Seq(GSE67343)/Homer | 1e-7 | -1.757e+01 |
| 5 | | GGAAGTGAAAGC | PU.1:IRF8(ETS:IRF)/pDC-Irf8-ChIP-Seq(GSE66899)/Homer | 1e-6 | -1.423e+01 |

**B**

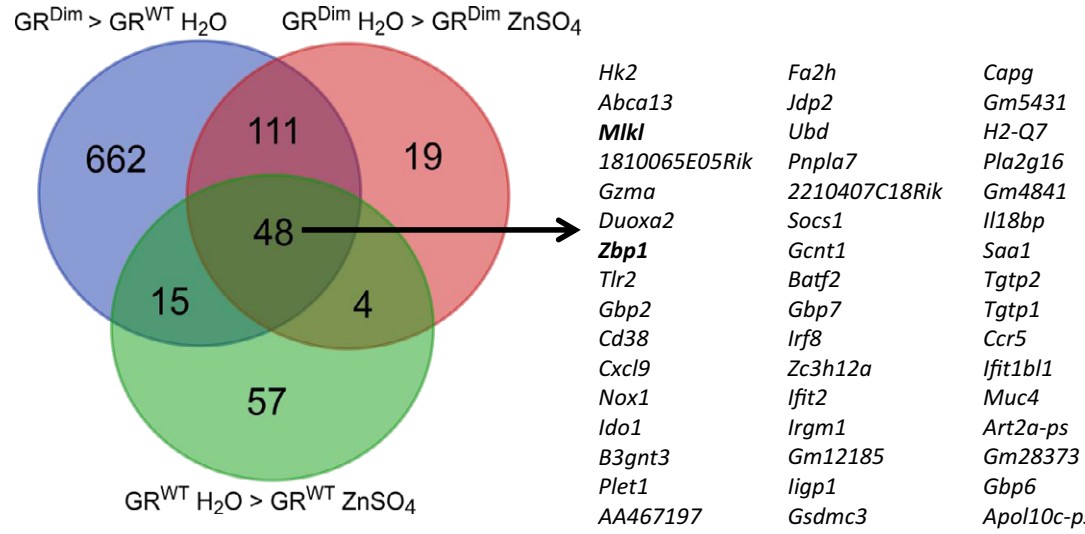

GR<sup>Dim</sup> > GR<sup>WT</sup> H₂O — GR<sup>Dim</sup> H₂O > GR<sup>Dim</sup> ZnSO₄ — GR<sup>WT</sup> H₂O > GR<sup>WT</sup> ZnSO₄

662 — 111 — 19 — 48 — 15 — 4 — 57

| | | |
|---|---|---|
| Hk2 | Fa2h | Capg |
| Abca13 | Jdp2 | Gm5431 |
| **Mlkl** | Ubd | H2-Q7 |
| 1810065E05Rik | Pnpla7 | Pla2g16 |
| Gzma | 2210407C18Rik | Gm4841 |
| Duoxa2 | Socs1 | Il18bp |
| **Zbp1** | Gcnt1 | Saa1 |
| Tlr2 | Batf2 | Tgtp2 |
| Gbp2 | Gbp7 | Tgtp1 |
| Cd38 | Irf8 | Ccr5 |
| Cxcl9 | Zc3h12a | Ifit1bl1 |
| Nox1 | Ifit2 | Muc4 |
| Ido1 | Irgm1 | Art2a-ps |
| B3gnt3 | Gm12185 | Gm28373 |
| Plet1 | Iigp1 | Gbp6 |
| AA467197 | Gsdmc3 | Apol10c-ps |

**C**

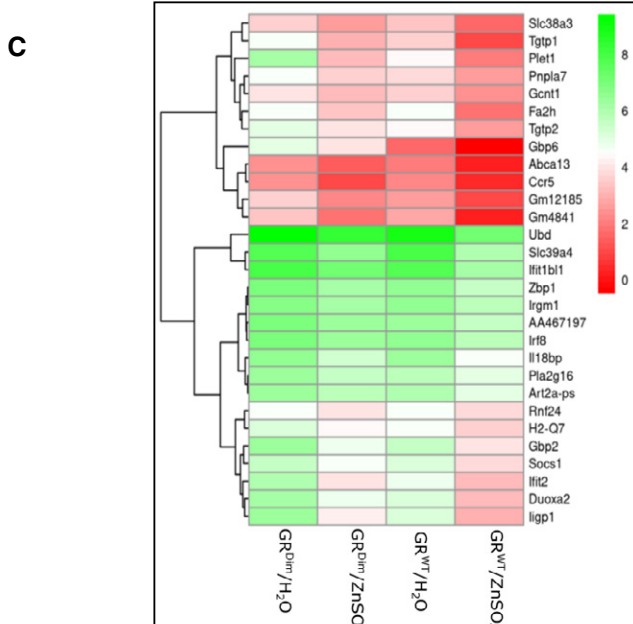

**Figure 3.**

**Figure 3. Transcriptional downregulation of ISRE/IRF genes in the intestinal epithelium by zinc part 1.**

RNA-seq experiment of GR$^{WT}$ and GR$^{Dim}$ mice which received normal drinking water or drinking water containing 25 mM ZnSO$_4$ for 1 week. Ileum was sampled and RNA-seq performed (N = 5–6 per group). Genes downregulated by zinc compared to water control animals (LFC < 0.8 and P < 0.05) were considered.

A  HOMER transcription factor motif analysis of genes downregulated by zinc in GR$^{WT}$ (left panel) and in GR$^{Dim}$ mice (right panel).

B  Overlap between the collection of genes, in blue, upregulated in GR$^{Dim}$ mice (water) compared to GR$^{WT}$ mice (water), in pink, genes downregulated by zinc in GR$^{Dim}$ mice and, in green, the genes downregulated by zinc in GR$^{WT}$ mice. The core of these three collections contains 48 genes which are listed.

C  Heat map of a selection of the 48 genes of Fig 3B.

Based on the lack of protection by zinc in GR$^{Dim}$ mice, a genome-wide study of protective pathways was initiated by treating GR$^{WT}$ as well as GR$^{Dim}$ mice with control drinking water or with 25 mM ZnSO$_4$ followed by a bulk RNA-seq on ileum samples (n = 5–6 each group). In the RNA-seq datasets, we studied the numbers of genes induced or reduced by zinc in GR$^{WT}$ and GR$^{Dim}$ mice. LFC of 0.8 and FDR of 0.05 were used as cut-off values. In GR$^{WT}$ animals, we found 378 transcripts induced and 88 reduced by zinc, and in GR$^{Dim}$ mice, respectively, 136 and 102 transcripts. Motifs enriched in the promotors of zinc-downregulated transcripts, surprisingly, were virtually exclusively interferon-stimulated response element (ISRE) and interferon regulatory factor (IRF)-binding sites, in GR$^{WT}$ and in GR$^{Dim}$. In both genotypes, ISRE, IRF1, IRF2, IRF3, and IRF8 formed the top-5 elements (Fig 3A). On the RNA-seq level, it is interesting to note that the majority of genes downregulated by zinc in GR$^{Dim}$ mice (the pink group in Fig 3B; 159/182) are genes whose basal expression is increased in these GR$^{Dim}$ mice, as compared to GR$^{WT}$ mice (the blue group in Fig 3B). If we also consider the genes which are downregulated by zinc in GR$^{WT}$ mice, then we find a core of 48 genes, which are listed. A gene set enrichment analysis on these 48 genes with significant enrichments (5% level, after multiple testing correction) was obtained using gene function lists from BioPlanet, WikiPathways, and KEGG. We found enrichments for type II interferon signaling (interferon-gamma) [BioPlanet and WikiPathways], interferon alpha/beta signaling [BioPlanet], interferon-gamma signaling pathway [BioPlanet], and toxoplasmosis [KEGG]. Enrichments were done using the Enrichr webtool. The expression levels in GR$^{WT}$ and GR$^{Dim}$ mice, and the impact of zinc, are illustrated in the heat map, to illustrate these results (Fig 3C).

Based on the finding that IECs of GR$^{Dim}$ mice chronically display high levels of ISRE/IRF genes and are extremely sensitive for TNF-induced Paneth cell death (Ballegeer et al, 2018), we hypothesize that these IECs are continuously stimulated by the microbiota to express ISRE/IRF genes and that GCs and GR (as homodimers) prevent this response by repressing Stat1 gene expression (Ballegeer et al, 2018). Since several ISRE/IRF genes encode proteins involved in necroptotic cell death (Ripk3, Mlkl, Zbp1), the increased expression of these genes by loss of GCs or GR (dimerization) may sensitize for TNF-induced necroptosis. The repression of the basal ISRE/IRF genes by zinc occurs in GR$^{WT}$ and GR$^{Dim}$ alike, suggesting that this repression is independent of GR, and happens at the level of the microbiota, the major ISRE/IRF-stimulating activator. When studying the expression of key ISRE/IRF genes (Mlkl and Zbp1) by qPCR in ileum of GR$^{WT}$ and GR$^{Dim}$ and the impact of zinc, we indeed observed an equal effect of zinc in both groups, but since these genes are higher expressed in GR$^{Dim}$ mice, the reduction by zinc remains incomplete, as shown in Fig 4A and B.

A subset of the ISRE/IRF genes shown in Fig 3B and C were measured by qPCR in other mutant mice exhibiting GC or GR

defects and in which zinc did not confer protection and were found to be increased in expression: This was the case in Adx mice and GR$^{VillKO}$ mice (Appendix Fig S5). We also studied ISRE/IRF genes in germ-free mice, STAT1$^{-/-}$ mice as well as several TLR-deficient mice (TLR2, TLR4, TLR5, and TLR9) and found that microbiota and STAT1 are essential in inducing these genes in ileum and that TLR4 plays a minor role, while the absence of the other TLRs had no repressive effects on the ISRE/IRF gene expression in ileum (Fig 4C–E).

In accordance with the hypothesis, other mice which were not protected by zinc, namely antibiotic-pretreated mice, had extremely low expression levels of ISRE/IRF genes in ileum and zinc no longer reduced their expression any further, as demonstrated in Fig 4F and G.

## The impact of zinc on TNF-induced crypt cell death and transport of microbes to the periphery

We studied the impact of TNF in mice treated with water or ZnSO$_4$ by light microscopy, immunohistochemistry, and transmission electron microscopy (TEM). Hematoxylin and eosin staining of sections of ileum taken 6 h after TNF injection clearly illustrate that TNF causes complete degranulation of Paneth cell granules, the contents of which appear to be smeared over the crypts and villi. Zinc conferred an obvious protection (Fig 5A). When slides were stained with an anti-MMP7 antibody, the specificity of which is illustrated in Appendix Fig S7, a specific Paneth cell granule component (Wilson et al, 1999), TNF led to weaker, more diffuse signals compared to controls and, again, zinc caused considerable protection (Fig 5B). We then studied the effects of TNF injection in mice on Paneth cells via TEM, and the impact of zinc thereon (Fig 5C). The pictures display typical examples and reveal that (i) untreated mice or Zn-treated mice show normal numbers of granules, normal endoplasmic reticulum (ER) structure, normal lumen size, and no cells extruded, no small apoptotic bodies; (ii) the devastating effects of TNF on Paneth cells as reflected by cell swelling, loss of granules, dark and swollen ER, many big apoptotic bodies, big lysosomes toward the lumen, extrusion of cells into the lumen, and loss of contact with the basal membrane; (iii) we observe that Zn is able to confer protection against these features to some extent, and the general morphology of the Paneth cells is in between control and TNF situation. Finally, we performed qPCR on ileum samples to detect Paneth cell-specific markers, the decline of which being a readout of less PCs, as described in the literature (Van Looveren et al, 2020). The expression of Defa6 and Lyz1, two Paneth cell markers, is shown in Fig 5D and E, and the data confirm that PC numbers decline by TNF, but significantly less by Zn treatment. Measuring Ripk3 by qPCR, on the other hand, confirms that necroptosis is activated by TNF, but less in zinc-pretreated mice (Fig 5F).

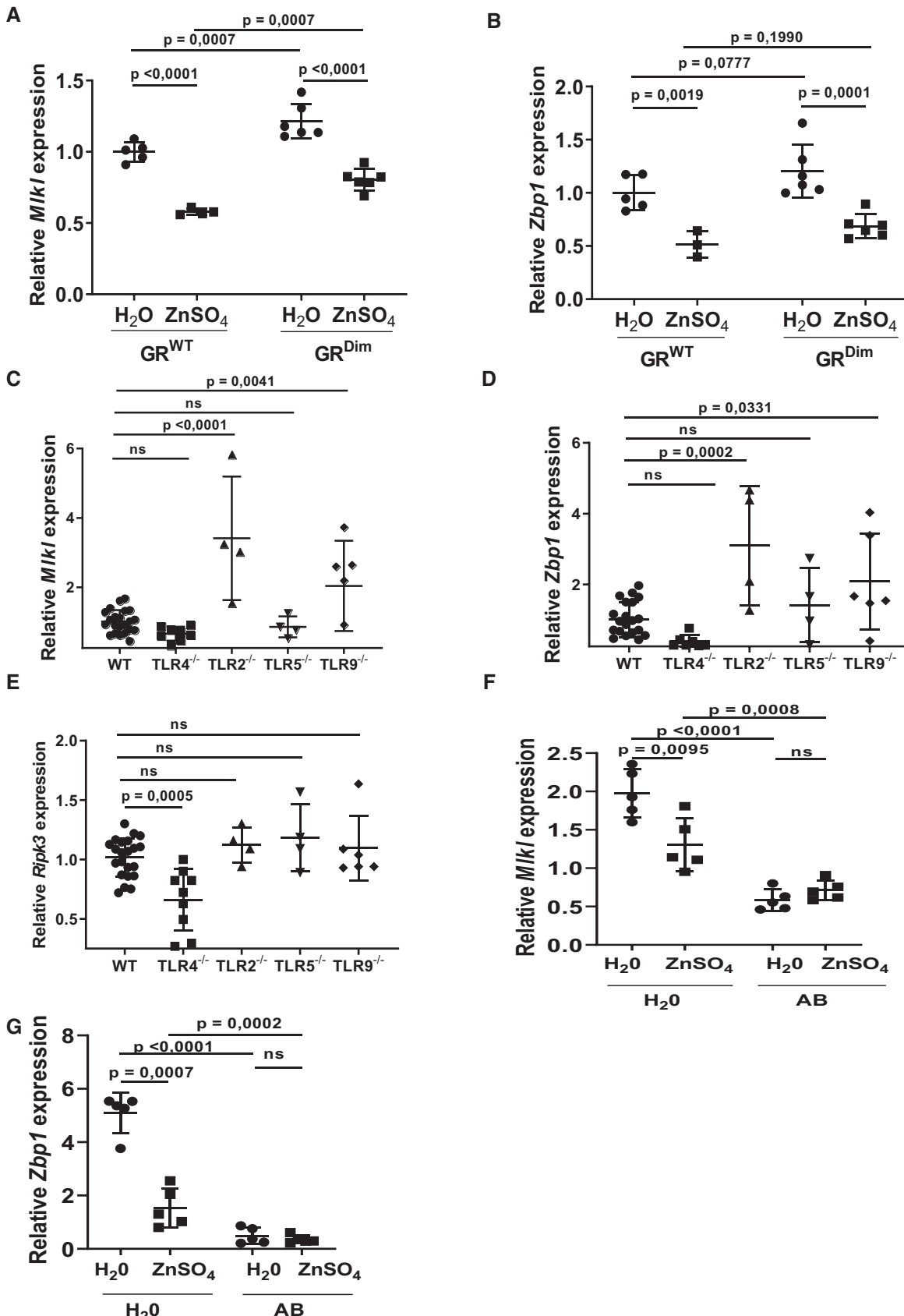

**Figure 4.**

◀

Figure 4.    Transcriptional downregulation of ISRE/IRF genes in the intestinal epithelium by zinc part 2.

A, B    Expression levels (RT–qPCR data) of *Mlkl* and *Zbp1*, from the collection of 48, in ileum biopsy samples, showing the reducing effects of zinc in both GR$^{WT}$ and GR$^{Dim}$ mice, and the higher expression in GR$^{Dim}$ compared to GR$^{WT}$ ($N$ = 4–6/group).

C–E    RT–qPCR data of *Mlkl*, *Zbp1*, and *Ripk3*, in ileum biopsy samples of C57BL/6J mice or different TLR full knockout mice. ($N$ = 4/23 per group).

F, G    RT–qPCR data of *Mlkl* and *Zbp1*, in ileum biopsy samples of C57BL/6J mice treated with normal drinking water or 25 mM ZnSO$_4$ in the drinking water for 1 week, either or not pretreated with antibiotics (AB) for 3 weeks ($N$ = 5 per group).

Data information: Data are shown as mean ± SD. In (A, B, F, G), $P$-values were analyzed with a two-way ANOVA test and in (C–E) with a one-way ANOVA test.

Since TNF-induced crypt damage was reported to cause evasion of (aerobic) microbes from the ileal lumen to peripheral organs (Van Hauwermeiren *et al*, 2015), causing there a bacterial infection that contributes to the lethal effect of TNF, we studied if zinc protection was associated with reductions in bacterial contamination of spleens and mesenteric lymph nodes, 18 h after TNF injection. Lysates of these organs were prepared and plated on TSA bacterial plates and colonies grown for 24 h, counted, picked, and at least 100 colonies identified for each condition by MALDI-TOF. TNF led to the growth of colonies in both organs, but significantly less when mice were pretreated with zinc (Fig 6A and B). Interestingly, zinc caused a reduction in the amount of bacterial species: In control mice, TNF causes the appearance of 4 dominant species on the TSA plates (*Staphylococcus nepalensis*, *Staphylococcus sciuri*, *Escherichia coli*, and *Enterococcus faecalis*), while zinc-pretreated mice displayed only the latter bacteria in their organs (Fig 6A and B) upon TNF injection. We therefore considered it conceivable that TNF causes Paneth cell death, leading to local crypt permeability of the intestinal epithelium, leading to bacterial evasion. We showed that zinc protects against Paneth cell damage, possibly allowing significantly less species and amounts of bacteria to evade to the spleen and MLNs. When intestinal permeability, measured 8 h after TNF injection by FITC-dextran, however, significant induction by TNF was observed, but no inhibition by Zn (Appendix Fig S6). Although the contribution of TNF-induced Paneth cell death in the degree of FITC-dextran-measured permeability assay is not known, the reduction in amount of colonizing bacterial species appears not to be directly determined by the general permeability of the gut, but by the composition of the ileum flora. This was determined by plating out ileum contents of mice, kept on normal drinking water ($n$ = 5), and mice ($n$ = 5) kept 1 week on 25 mM ZnSO$_4$. *Grosso modo* equal amounts (about 200 mg/mouse) of ileum slurry were collected in PBS and plated on TSA plates. Slightly (10%) more colonies appeared on the plates of the zinc-treated mice, but after picking, randomly 110 and 102 colonies from plates of H$_2$0-treated mice and ZnSO$_4$-treated mice resp. and determination by MALDI-TOF, a significant impact of Zn was observed, since the abundant *Staphylococcus* species (*S. sciuri*, *S. nepalensis*, and the far less abundant *S. saprophyticus*) were entirely replaced by *E. coli* and *Enterococcus faecalis* (Fig 6C and D).

To understand the mechanism of the effect of zinc on the gut microbial communities, we used two techniques. First, we studied the four major bacterial species that were found to colonize the spleen and the MLNs after TNF injection to study direct effects of zinc. Plating out of mice-derived isolates on TSA plates containing increasing concentrations of ZnCl$_2$ revealed that the growth of *Staphylococcus nepalensis* and *Staphylococcus sciuri* (both *Firmicutes*) is severely limited by zinc, leaving no live bacteria in 2.5 mM ZnCl$_2$, while *Enterococcus faecalis* (*Firmicutes*) is fully alive even on

the highest concentration of 5 mM Zn (Fig 6E). *Escherichia coli* (*Proteobacteria*) displays an intermediate direct Zn cytotoxic effect. These data are compatible with a direct antibiotic effect of zinc on some bacterial species/taxa, but not on others, explaining the impact of zinc on the ileum gut microbes identified by MALDI-TOF and those appearing after TNF challenge in MLNs and spleens, identified by MALDI-TOF.

Based on the hypothesis that ileum bacteria induce an ISG response in Paneth cells, sensitizing them to TNF-induced cell death, we decided to study whether mice, protected against TNF by zinc treatment, can be re-sensitized to TNF by transplanting them with *Staphylococci*. Mice were treated with H$_2$0 or ZnSO$_4$ for 7 days, after which the ZnSO$_4$ was removed and replaced by H$_2$0. 12 h and 24 h later, the mice were transplanted with $10^8$ *S. sciuri* or $10^8$ *S. nepalensis*, and 3 h later, a lethal dose of TNF was injected, and 24 h later, body temperature and lethality recorded (Fig 6F). The data suggest that Zn induces protection against TNF lethal shock, but that this Zn protection is reversed by transplanting mice with these bacterial species. To confirm that the lethal response of these transplanted mice is related to expression of ISRE/IRF genes in the ileum at the moment of challenge, we performed qPCR of two such genes, *Zbp1* and *Stat1*, and observed the repression by zinc and the reversal by the bacterial gavage. *Mt2*, a zinc-activated gene, we measured as a positive control of zinc treatment (Fig 6G–I).

## Impact of zinc on the (fecal) gut microbiome by 16S rRNA typing

To investigate whether the impact of zinc on microbiota composition, like the reduction in ISRE/IRF genes, is independent of GR and is a direct effect of zinc on the microbes, we first compared the composition of the fecal microbiota of GR$^{WT}$ mice ($n$ = 40) and GR$^{Dim}$ mice ($n$ = 40) by 16S rRNA sequencing on feces. As shown in Appendix Fig S8, there were slightly more *Bacteroidetes* in GR$^{WT}$ than in GR$^{Dim}$ and less *Firmicutes*, but none of the differences, at any level from phylum to genus level, were statistically significant.

We then investigated the effect of zinc on the gut microbiome and performed a 16S rRNA gene sequencing on feces sampled from 20 control and 20 ZnSO$_4$-treated C57BL/6J mice. Taxonomy was assigned by using the Silva database, and analysis was performed by the use of Qiime1. The fecal microbial community diversity between both groups (α-diversity) was compared by means of Shannon-Wiener rarefaction curves. As suggested by Fig 6D and E, the fecal microbiome of zinc-treated mice was indeed markedly decreased in α-diversity, pointing to a reduction in fecal bacterial richness and diversity ($P$ = 0.001; Appendix Fig S9A). The individual variation in both treatment groups (β-diversity) was assessed with Bray–Curtis dissimilarity (Appendix Fig S9B). Individual variation was significantly greater between control and zinc treatment groups than within the groups ($P$ = 0.001). Individual variation

within groups was not statistically different ($P = 0.146$ within the zinc group, $P = 0.118$ within the control group). Taxonomical differences on the phylum level between both groups are presented in Appendix Fig S10A. Quite strong increases in the abundance of

*Actinobacteria* (2% to 3.8%) and *Bacteroidetes* (46.2% to 65.7%) and decreases in *Firmicutes* (42.2% to 24%) and *Proteobacteria* (8.1% to 2.1%) are observed in the feces of zinc-treated mice. The decrease in *Firmicutes* may involve direct effects of zinc on

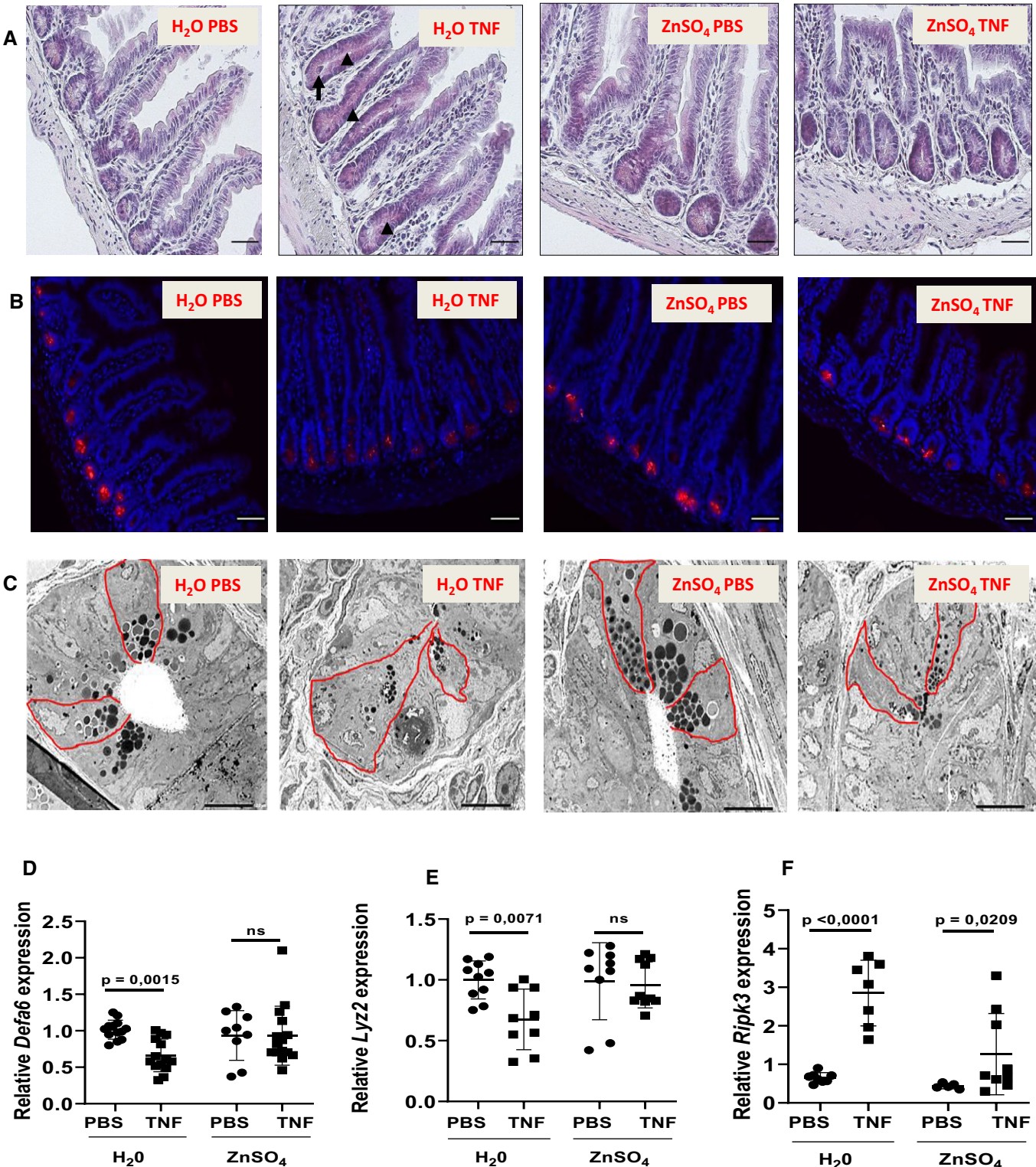

**Figure 5.**

◄

**Figure 5. Zinc protects Paneth cells from TNF-induced cell death and prevents TNF-induced bacterial invasion of spleens and mesenteric lymph nodes.**

A, B C57BL/6J mice received control or 25 mM $ZnSO_4$ drinking water for 7 days and were injected i.p. with 50 μg TNF, solved in 200 μl sterile PBS, per 20 g bodyweight or with PBS only. Six hours later, sections through ileum were made and stained with (A) hematoxylin & eosin and MMP7, a Paneth cell marker. Pictures at magnification of 20× were taken, and one representative picture of each condition is shown. Arrows point to crypts and their (ab)normal occurrence (see text). Scale bars: 50 μm.

C Transmission electron microscopy (TEM) images of ileum crypts show protective effects of zinc therapy on the level of TNF-induced effects on Paneth cells. C57BL/6J mice ($N = 2$ /group) were put on normal drinking water or on 25 mM $ZnSO_4$ water for 7 days. Mice were given 50 μg TNF or PBS, and 8 h later, ileum samples were taken and prepared for TEM. Figure shows ileal crypts at a magnification of 1,000×. In each of the four conditions, two Paneth cells are selected in red area (see text).

D–F RT–qPCR measurement of mRNA levels of *Defa6* and *Lyz2*, two Paneth cell markers, and *Ripk3*, a TNF-induced cell death gene, in ileum of the experiment in (A).

Data information: Data are shown as mean ± SD. (D) $N = 10/15$ per group, (E) $N = 10$ per group, (F) $N = 5/6$ per group. *P*-values were analyzed with a two-way ANOVA test.

*Staphylococci*, while the latter decrease in *Proteobacteria* is interesting since this phylum includes a wide variety of potentially pathogenic bacteria (e.g., *Salmonella*), and moreover, an increase in *Proteobacteria* was noticed in clinical SIRS and sepsis studies (Zaborin *et al*, 2014; Lankelma *et al*, 2017). To discriminate between bacterial genera characteristic for control feces or zinc-feces a linear discriminant analysis (LDA) was performed (Appendix Fig S10B). Interestingly, several genera previously identified as opportunistic pathogens and linked to (infectious) diseases declined upon zinc treatment. These include the *Desulfovibrio*, *Alistipes*, *Coprococcus*, and other genera (Wensinck *et al*, 1983; Gulletta *et al*, 1986; Kasten *et al*, 1992; Bernard *et al*, 1994; Goldstein *et al*, 2003; Lau *et al*, 2006; Verstreken *et al*, 2012; Sydenham *et al*, 2014; Maharshak *et al*, 2018).

**Therapeutic consequence of the findings and conclusion**

In previous work (Ballegeer *et al*, 2018), we have shown that GCs/GR control microbiota induced ISRE gene expression in Paneth cells, by controlling STAT1 expression and activity, while here, we show that zinc directly modulates the gut microbiome, so that less ISRE genes are induced in Paneth cells. Thus, zinc and GCs hit the same intestinal TNF-induced cell death pathway to protect against TNF-induced lethality, but zinc acts more upstream in that pathway. Hence, zinc and GCs might be expected to strengthen the effects of one another. To test this, we injected mice with a high dose of TNF and found that suboptimal doses of Dex and zinc, each failing to cause a full protection on their own, do so when combined in a single treatment (Fig 7A). Since both protective mechanisms converge at STAT1, we studied if STAT1$^{-/-}$ mice phenocopy this robust TNF protection, which indeed they do (Fig 7A). Furthermore, when studying the ileum expression of ISRE genes of interest in TNF-induced cell death (*Mlkl* and *Zbp1*) 6 h after TNF injection, in mice pretreated with DEX (−1 h) and put on water or $ZnSO_4$ during a week, it is observed that TNF induces these genes, that DEX represses this induction, and that $ZnSO_4$ reduces the expression of these genes in all groups, including the DEX-treated group, leading to significant additive effects of $ZnSO_4$ and DEX (Fig 7B–D). Such effect is also observed at the level of *Stat1* expression.

In conclusion, we believe that our data suggest that zinc modulates the composition of the gut microbial communities, in a direct way, by killing certain bacterial species, such as Staphylococci, leading to reduced microbial pressure on IECs such as Paneth cells, and to reduced ISRE/IRF gene expression in these cells. Since some of these genes are coding for proteins involved in necroptotic cell death, zinc reduces vulnerability to TNF-induced cell death, thereby protecting against bacterial colonization from ileum to spleen and MLNs, and lethality (Fig 7E).

## Discussion

Since TNF has been shown to play a major role in numerous inflammatory and infectious diseases (Chen *et al*, 2007; Bhattacharyya *et al*, 2011), knowledge of the precise impact of TNF-induced inflammation and cell death on the cellular and pathophysiological level are of major importance (Tang *et al*, 2019). Clinical trials using TNF infusion in healthy human volunteers have shown that, besides drop in blood pressure, the impact of TNF on intestinal homeostasis (diarrhea, damage, pain, necrosis) was the major dose-limiting toxicity (Spriggs *et al*, 1988). It has been demonstrated that TNF-induced lethal SIRS in mice, which is used as a model system for sepsis but also for inflammatory bowel disease, is strongly mediated by cell death in the IECs (Piguet *et al*, 1998; Van Hauwermeiren *et al*, 2013, 2015) and is associated by efflux of gut microbes into the spleen and MLNs. This latter phenomenon is important and explains why broad-spectrum antibiotics can protect mice against a lethal TNF challenge (Van Hauwermeiren *et al*, 2015). TNF inhibitors have proven to block inflammatory bowel disease (IBD) in Crohn's disease, which proves the role of TNF in this inflammatory disease (Guo *et al*, 2010). GCs efficiently prevent inflammation and are applied in IBD (Danese & Peyrin-Biroulet, 2014). They are also very efficient in preventing lethality in this TNF model by a mechanism that is based on inhibition of gene expression of ISRE/IRF genes in IECs, genes which are induced by the gut microbiota and some of which (*Mlkl*, *Zbp1*) sensitize for TNF-induced necroptosis (Ballegeer *et al*, 2018).

Zinc is the second most abundant trace element in the human body and is important for health as a structural, functional, and catalytic cofactor of approximately 10% of all human proteins (Andreini *et al*, 2006). As a consequence, zinc is involved in many biological pathways and zinc deficiency is associated with several clinical features, mainly gastrointestinal, such as diarrhea and increased susceptibility to infections, SIRS, and sepsis (Livingstone, 2015). Zinc administration protects against TNF-induced lethal SIRS, as well as in other models of SIRS, sepsis, and infection. Zinc has been applied in the combat against diarrhea in newborn children and in piglets, but the precise mechanism underlaying these zinc effects are insufficiently known (Waelput

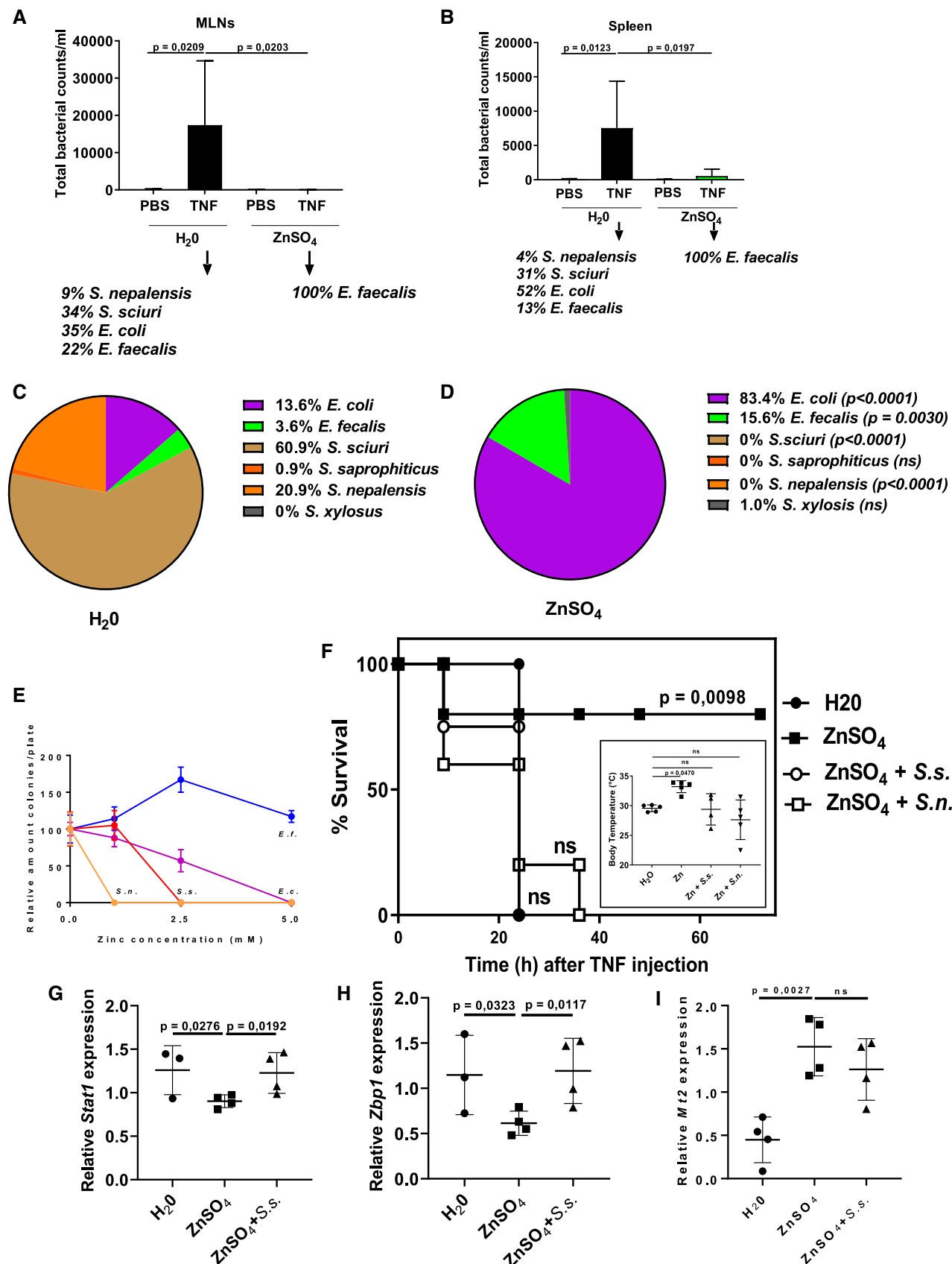

Figure 6.

◄

**Figure 6. Zinc modulates bacterial colonization and composition as a mechanism to protect against TNF.**

A, B  C57BL/6J mice received control or 25 mM ZnSO$_4$ drinking water for 7 days and were injected i.p. with 50 μg TNF, solved in 200 μl sterile PBS, per 20 g bodyweight or with PBS only (N = 5). 18 h after TNF challenge, spleen and MLNs were isolated and lysates prepared and plated on TSA plates. Colonies were counted and expressed as bacterial colonies per ml of lysate. 40 colonies per plate were prepared for MALDI-TOF identification, and the relative amounts of bacterial species found on the plates are mentioned underneath the figure. Data are shown as means ± SD. P-value was analyzed by means of Student's t-test (unpaired, two-tailed).

C, D  C57BL/6J mice received control or 25 mM ZnSO$_4$ drinking water for 7 days. Then, semi-solid ileum slurry was isolated, weighed, mixed, diluted, and normalized and was plated on TSA plates. After randomly picking 110 and 102 colonies from plates of H$_2$O-treated mice and ZnSO$_4$-treated mice resp., their identity was determined by MALDI-TOF and expressed in a pie diagram. Chi-square tests were performed.

E     Impact of zinc concentration in bacterial TSA plates on the growth of colonies of four bacterial species commonly found in spleen and MLNs of mice injected with TNF (see A, B). Pure colonies of *Staphylococcus nepalensis* (orange), *Staphylococcus sciuri* (red), *Eschericchia coli* (purple), and *Enterococcus faecalis* (blue) were grown in LB medium, then diluted, and plated on TSA plates containing increasing concentrations of ZnCL$_2$. 24 h later, the amounts of colonies were counted, and the amounts found in the 0 mM ZnCl$_2$ condition expressed as 100%. Data are shown as mean ± SD pooled from three independent experiments.

F–I   Monocolonization experiment. Mice (N = 5 all groups) were treated with H$_2$O or ZnSO$_4$ for 7 days, after which the ZnSO$_4$ was removed and replaced by H$_2$O. 12 h and 24 h later, the mice were transplanted with PBS, 10$^8$ *Staphylococcus sciuri* or 10$^8$ *S. nepalensis* (in 100 ul), and 3 h later, a lethal dose of TNF (35 μg/mouse) was injected, and 24 h later body temperature (insert in F) and lethality (F) recorded and estimated by one-way ANOVA and chi-square tests, resp. Survival significance was tested toward the H$_2$O control group. In a second experiment (G–I; N = 4 all groups) with *S. sciuri* colonization, *Zbp1* and *Stat1*, two key ISRE genes, and *Mt2*, a Zn-induced gene, were measured in ileum biopsies by RT–qPCR.

Data information: Data are shown as mean ± SD. In (A), (B), and (F), P-values were analyzed with one-way ANOVA test and in (G-I) by means of Student's t-test (unpaired, one-tailed). For the survival curve in (F), P-value was analyzed with a chi-square test.

---

et al, 2001; Van Molle et al, 2007; Nowak et al, 2012; Wessels & Cousins, 2015; Ganatra et al, 2017). Since the health-promoting effects of zinc are well-known, the protective effects of zinc in the TNF SIRS model form an ideal opportunity to study mechanistic insights that lay behind the protective nature of zinc. This may lead to an optimization or replacement of zinc as a therapeutic approach, since zinc is a heavy metal causing fear for environmental contamination.

We here describe that zinc protects mice very well against TNF-induced SIRS but is unable to confer protection when the GC/GR axis is severely disturbed. By RNA-seq experiments, we decided to focus further on the ileum, although the GR$^{VillKO}$ data may suggest that zinc's protective effects could also be mediated by other cells in addition. The RNA-seq data exclude that zinc communicates directly with GR and influences its transcriptional efficiency. Because lethality in the TNF model is associated with severe cell death in the small intestinal crypts, notably in the Paneth cells, and since this TNF phenomenon on Paneth cells depends on sufficient expression of ISRE/IRF genes, induced by the gut microbiome (Ballegeer et al, 2018), we studied whether a recently recognized interaction between GR and zinc may be playing a role. Guo et al have shown that the *Slc30a2* gene is regulated by a collaboration between GR and the major zinc-responding transcription factor MTF1 (Guo et al, 2010). The *Slc30a2* gene encodes a major zinc transporter, ZnT2, which pumps zinc from the cytoplasm into the secretory granules in Paneth cell, where zinc supposedly helps in the maturation of beta-defensins via stimulation of MMP7, the major protease involved, as elegantly shown by Wilson et al (Wilson et al, 1999). However, in

our model, by using newly generated Slc30a2 knockout mice, MTF1-deficient mice, and MMP7 knockout mice, we found no support for the involvement of the GC/GR-MTF1-ZnT2-MMP7 axis in the zinc protective effect.

Instead, we found that zinc no longer protects mice against TNF when gut microbes have been killed by classical antibiotics or have been absent by raising mice in germ-free (GF) conditions. Both these interventions led to a drastic reduction in the expression of the ISRE/IRF-dependent genes in IEC cells, and the antibiotics already sufficed to protect against TNF, as shown before (Van Hauwer-meiren et al, 2015). Remarkable, though GF mice had no microbes in their gut, they were not basically protected against TNF, maybe because of the immature status of their innate immune defense systems (Bayer et al, 2019). Taken together with the strongly increased expression of these ISRE/IRF-dependent genes in GC/GR-deficient animals, it is reasonable to assume that the protective effect of zinc is based on the reduction in these ISRE/IRF-dependent genes in a way that involves the microbiota and Paneth cell's cell death.

Our RNA-seq study following zinc administration in IECs clearly showed that zinc reduces the basal ISRE/IRF signature. It is the major effect of zinc in the gut. Interestingly, although GR$^{Dim}$ mice have basic high ISRE/IRF-dependent gene expression in the ileum, these mice with defective GR dimerization pathway respond well to zinc in terms of downregulation of ISRE/IRF-dependent gene expression when looking at responses on the genome-wide level. These data illustrate that the zinc effect is in fact independent from GR, but has no protective result, because the threshold of the ISRE/IRF-

---

**Figure 7. Additive protection of zinc and dexamethasone against TNF-induced lethal inflammation.**

A     C57BL/6J mice received 25 mM ZnSO$_4$ or control water for 7 days. Mice were injected i.p. with 100, 200, or 500 μg dexamethasone (Dex, rapidexon), solved in sterile PBS, 30 min before a 50 μg TNF/20 g bodyweight i.p. challenge. STAT1$^{+/+}$ and STAT1$^{-/-}$ mice were challenged with equal TNF dose, as controls. No mice died later than 120 h after TNF injection. Data show combined results of two experiments.

B–D   Mice (N = 4 per group) treated as in (A) and 6 h after challenge, ileum was taken and RT–qPCR performed to study gene expression of cell death ISRE genes *Mlkl* and *Zbp1* as well as *Stat1*.

E     Hypothesis of the mechanism of interference of zinc in TNF-induced cell death in ileum (Paneth) cells. See text for more details.

Data information: For the survival curve in (A), P-value was analyzed with a chi-square test and compared to the PBS group. In (B-D), pairwise analysis between groups with or without zinc was performed using Student's t-test (unpaired, one-tailed).

►

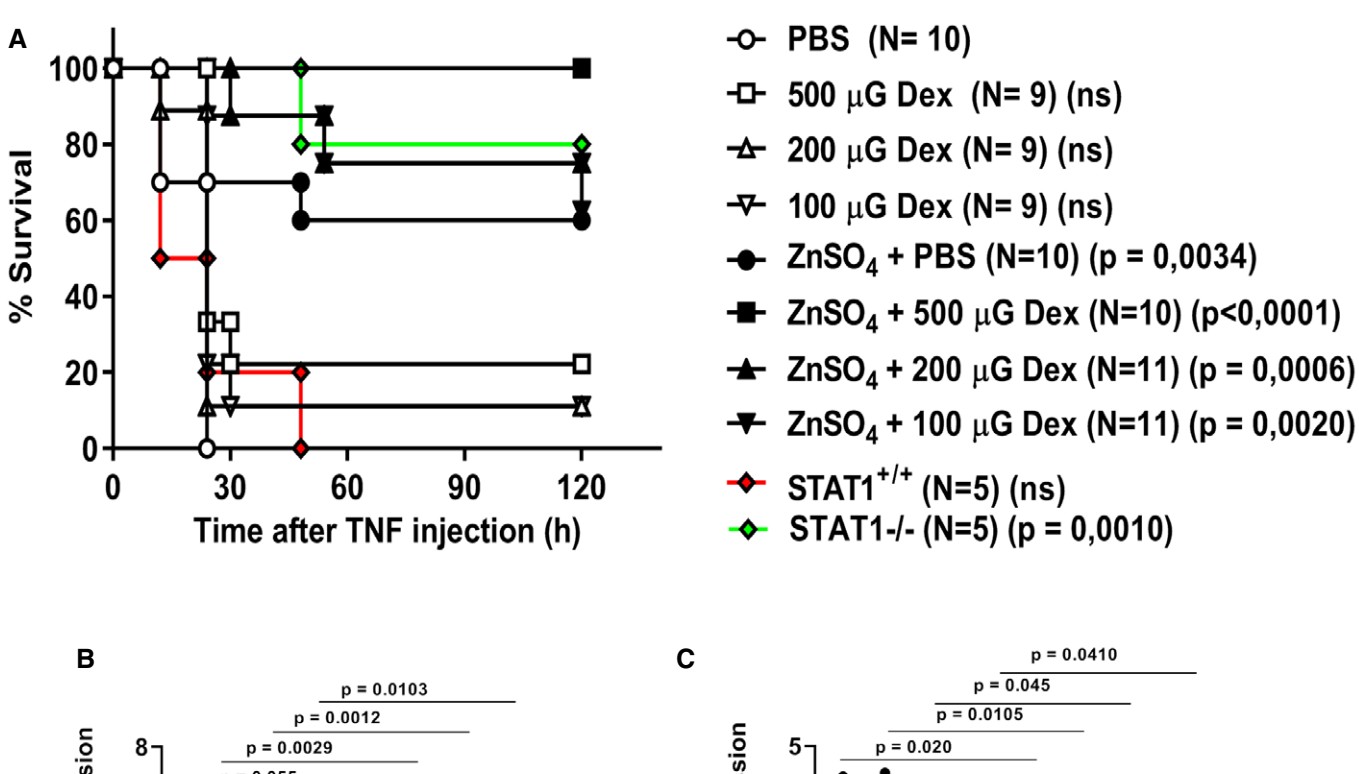

Figure 7.

dependent gene expression in these GR$^{Dim}$, but also GR$^{VillKO}$ and Adx mice, are too high. Increasing the dose of zinc in the drinking water from 25 to 75 mM still caused no protection in GR$^{Dim}$ mice, actually caused visible discomfort in the mice. Hence, zinc protects against TNF by lowering ISRE/IRF genes in the ileum, but not in GC/GR-deficient mice, because these genes are too high, and not in antibiotics and GF mice, because their ISRE/IRF genes are already too low.

Which are the key bacterial species and which pathways lead to ISRE/IRF gene expression, and how zinc modulates these was then addressed. It is known that zinc has microbiota modulating effects. This has been suggested in the literature, for example, in chickens (Mayneris-Perxachs et al, 2016), pigs (Xia et al, 2017) and mice (Li et al, 2016; Zackular et al, 2016). This was also underscored following our 16S rRNA gene sequencing. We found that the amounts of bacteria, after 1 week of zinc treatment, remain the same, but that the diversity and virulence decrease by zinc. Strong reductions in Proteobacteria and Firmicutes are obvious effects of zinc. Since TNF injection leads to the dissemination of gut aerobic bacteria in MLNs and gut, and since zinc strongly reduced this phenomenon, we cultured aerobic bacteria on plates from ileum isolates and found that zinc-treated mice had obviously lost the abundant Staphylococcus sciuri and Staphylococcus nepalensis species. Since these species, more than others, appear to be killed or stopped in their growth by zinc, in a direct way, it is conceivable that (i) this effect causes their absence in the MLNs and spleen, (ii) causes less ISRE/IRF expression in the ileum, and (iii) therefore causes protection of Paneth cells against TNF-induced (ISRE/IRF-dependent) cell death. Direct proof for the role of these bacterial species was provided by monoculture transplantation studies, showing that these bacteria reverted the reduction in ISRE/IRF genes by zinc as well as the protection offered by zinc against TNF, in terms of hypothermia and death. S. sciuri is better understood than S. nepalensis and is an important human pathogen, found in endocarditis, peritonitis, septic shock, urinary tract infection, pelvic inflammatory disease, and wound infections (Chen et al, 2007). It has also been found in pigs and is considered as a potentially dangerous zoonotic agent (Chen et al, 2007). More details on how S. sciuri and S. nepalensis contribute to ISRE/IRF upregulation will have to be unfolded by later studies. Also, monoculture transplantation studies using other bacterial strains, which were not depleted by ZnSO$_4$, or which will be identified by microbiota sequencing studies rather than by culturing but will increase the specificity of these findings and make them more complete, in the future. We found that ablation of TLR-coding genes (Tlr2, Tlr4, Tlr5, Tlr9) had some impact on the basal expressions of ISRE/IRF genes, but more research is needed, since it is conceivable that other pathogen sensors, such as cGAS/STING or NLRs, play a role (Zhang et al, 2020). As an alternative explanation of the impact of zinc on ileum ISRE/IRF gene expression, zinc may have a direct role by repressing IRF-related genes by binding to the newly identified zinc-finger IRF composite elements (Ochiai et al, 2018).

This study unfolds an interesting axis, which potentially leads to new insights and new therapeutic possibilities. Zinc reshapes the microbiome, by direct cytotoxic/cytostatic effects on certain bacteria, such as Staphylococci, leading to a less-diverse community that induces less ISRE/IRF-dependent gene expression. Since some of these ISRE/IRF genes, notably Mlkl and Zbp1, code for necroptotic proteins MLKL and ZBP1, a direct link between ISRE/IRF levels and sensitivity to TNF-induced necroptosis is strongly suggested by microscopy and electron microscopy studies. Interestingly, the most central node in regulation of these ISRE/IRF genes is STAT1, coded by Stat1. Our data suggest that zinc and GCs have additive protective effects against TNF-induced lethal SIRS. The protection of GCs was shown to be directed, to a significant degree, to direct inhibition of Stat1 gene expression (Ballegeer et al, 2018; Van Looveren et al, 2020), as STAT1 is a well-known mediator in TNF signaling and Stat1 a target of GR (Yarilina et al, 2008; Bhattacharyya et al, 2011). Our paper suggests that zinc reduces ISRE/IRF gene expression because of less signal transduction in ileal cells (including Paneth cells), which is also reflected in reduced Stat1 expression. We therefore believe that zinc and GC protection converge at Stat1 control. STAT1 is key in TNF signaling and lethal shock, as shown before (Yarilina et al, 2008; Ballegeer et al, 2018), and confirmed here by means of full STAT1 knockout mice. The details of the role of STAT1 in intestinal epithelium in the TNF model, but also in IBD and Crohn's disease, are being revealed (Seamons et al, 2018; Gunther et al, 2019), but await further detailed studies. Since administration of zinc and GCs together has such strong effects against TNF-induced lethality, this combination may be of utmost benefit for treating intestinal pathologies involving TNF, such as Crohn's disease.

# Materials and methods

### Mice

Female C57BL/6J and C57BL/6J Adx mice were obtained from Janvier (Le Genest-St.Isle, France). GR$^{Dim}$ breeding mice (on FVB/NJ background) were kindly provided by Dr. Jan Tuckermann (Ulm University, Germany) and bred as GR$^{Dim/+}$ intercrosses, yielding GR$^{Dim/Dim}$ (noted as GR$^{Dim}$) and GR$^{WT/WT}$, noted as GR$^{WT}$. GR$^{VillKO}$ mice were obtained by crossing GR$^{fl/fl}$ mice, obtained via Dr. Jan Tuckermann, with Villin-cre-transgenic mice, in a C57BL/6J background. GR$^{fl/fl}$ mice were used as control mice for GR$^{VillKO}$ mice, and cre-transgenic animals were not considered as controls. A20$^{VillKO}$ mice were obtained by crossing A20$^{fl/fl}$ mice, and generated and provided by Dr. Geert van Loo (IRC VIB, UGent, Belgium) with Villin-cre-transgenic mice, in a C57BL/6J background. MTF1$^{fl/fl}$ mice (on a C57BL/6J background) were obtained by Dr. Walter Schaffner (Zurich, Switzerland), and MMP-7$^{-/-}$ mice (C57BL/6J background) were from Dr. Carole Wilson (Seattle, USA, courtesy of Dr. Lynn Matrisian). STAT1$^{-/-}$ mice were purchased from the Jackson Laboratory. Slc30a2$^{-/-}$ mice were generated by CRISPR/Cas mutagenesis in 57BL/6J zygotes by Dr. Tino Hochepied in our institute (IRC, VIB, UGent, Belgium). All in-house breedings were maintained by heterozygous crossing. All TLR$^{-/-}$ mice were in a C57BL/6J background and kept in SPF animal house.

All mice were housed in light controlled (14-h light/10-h dark) and air-conditioned animal houses. Mice were kept in individually ventilated cages and received food and water ad libitum. When mice were treated with zinc, they were given 25 mM ZnSO$_4$*7H$_2$O (Merck, 7446-20-0) solved in ultrapure water for 7 days. The drinking water of Adx mice was supplemented with 0.9% NaCl.

C57BL/6J, C57BL/6J Adx, Slc30a2$^{-/-}$, MMP7$^{-/-}$, and A20$^{VillKO}$ mice were kept in specific pathogen-free (SPF) animal houses. GR$^{Dim}$, MTF1$^{VillKO}$, and GR$^{VillKO}$ mice were kept in a conventional animal house. The GF experiment of Fig 2 was performed in C57BL/6J mice in the GF facility of the Instituto Gulbenkian de Ciência (Lisbon, Portugal). The other GF experiments were done in the GF facility of VIB: A germ-free C57BL/6J colony is kept in the VIB department, in special isolators. Our C57BL/6J germ-free colonies are monitored for extraneous bacteria, fungi, and other pathogens. Male and female germ-free mice were transported to a BL2 room for immediate experimentation, at the age of 8 weeks old, when needed.

All mice were used at the age of 8–14 weeks. Experiments with mice from in-house breedings used both male and female mice, except for the RNA sequencing in GR$^{WT}$ and GR$^{Dim}$ mice where only female mice were used. All experiments were approved by the institutional ethics committee for animal welfare of the Faculty of Sciences, Ghent University, Belgium. The GF experiment of Fig 2 was performed in the GF facility of the Instituto Gulbenkian de Ciência (Lisbon, Portugal) in accordance with Portuguese regulations and approved by the Instituto Gulbenkian de Ciencia ethics committee and DGAV.

### Reagents

Dexamethasone (D-4902) and FD4 FITC-dextran (FD4-1G, MW 3,000–5,000 Da) were purchased at Sigma-Aldrich. Rapidexon (Eurovet) was used for *in vivo* experiments. Recombinant mouse TNF was expressed in and purified from *E. coli* in our department and contained no detectable endotoxin contamination (batch numbers CE140902 and SDB151216001, PSF VIB).

### Injections, monitoring, and blood and tissue isolations

Mice were injected i.p. or i.v. with an LD$_{100}$ TNF dose in a volume of 200 μl /20 g bodyweight. Injected doses are indicated in the figure legends and depend on the used TNF batch and the mouse strain. CS was isolated and prepared as described in Starr *et al* (2014). Rapidexon (500, 200, or 100 μg per 20 g mouse weight) was injected i.p. 30 min before TNF injection. TNF and rapidexon were diluted in endotoxin-free PBS (Gibco). Blood was sampled from the retro-orbital plexus during isoflurane sedation (Isoflo, Abbott animal health). Mice were killed by cervical dislocation for the isolation of IECs, ileum, cecal content, ileum slurry, spleen, mesenteric lymph nodes (MLNs), or liver samples. Samples for RNA isolations were stored in RNA later (Ambion). IEC samples for RNA isolations were prepared as follow: ± 5 cm of the distal ileum was dissected, flushed with ice-cold PBS, and incubated with lysis buffer (732 6802, Bio-Rad) supplemented with 2-mercaptoethanol on ice for 5 min, after which the inner part of the bowel was scraped out and snap-frozen in liquid nitrogen.

### Serum preparation, zinc, corticosterone measurements

To obtain mouse serum, blood samples were allowed to clot overnight at 4°C. The next day, the clot was removed and samples were centrifuged at 2,000 *g* for 4 min at 4°C. Serum samples were stored at −20°C. Zinc levels were measured via ICP-MS (Perkin-Elmer DRce) performed by the University Hospital of Ghent. Corticosterone levels were determined via the Coat-a-count rat corticosterone *in vitro* diagnostic test kit (TRKC1, Siemens Med. Sol.).

### RNA isolation, cDNA, and qPCR

Total RNA from IECs, ileum, and liver was isolated with the RNeasy Mini Kit (Qiagen). For RNA isolations of cells, TRIzol (Gibco, Life Technologies) and the InviTrap Spin Universal RNA Mini Kit (Invitek, Isogen Life Science) were used. Both RNA isolation kits were used according to the manufacturer's instructions. RNA concentration was measured with the NanoDrop 1000 (Thermo Fisher Scientific). Thousand nanogram RNA was used for cDNA synthesis (iScript Advanced cDNA Synthesis Kit, Bio-Rad). qPCR (LightCycler 480, Roche) was performed with the SensiFast SYBR No-ROX kit (Bioline) according to the manufacturer's instructions. The best performing housekeeping genes were determined by geNorm. For qPCR assessment of bacterial loads, the extracted DNA was quantified with the NanoDrop 1000 and diluted to 3 ng/μl. Three microlitre DNA was added to 7 μl mix (1.5 μl sterile nuclease-free water 5 μl SensiFast, 0.3 nmol Eubacteria forward + reverse primer). Primers used for qPCR are depicted in Table 1. Results are given as relative expression values normalized to the geometric mean of the housekeeping genes.

### RNA-seq of ileum and liver in C57BL/6J and GR$^{WT}$ and GR$^{Dim}$

RNA quality was checked with the Agilent RNA 6000 Pico Kit (Agilent Technologies). The RNA was used for creating an Illumina sequencing library using the Illumina TruSeqLT stranded RNA-seq library protocol (VIB Nucleomics Core), and single-end sequencing was done on an Illumina NextSeq 500. The obtained reads were mapped to the mouse (mm10) reference transcriptome/genome with hisat v2.0.4 (Kim *et al*, 2015). Gene-level read counts were obtained with the featureCounts software (part of the subread package) (Liao *et al*, 2014). Multimapping reads were excluded from the assignment. Differential gene expression was assessed with the DESeq2 package (Love *et al*, 2015), with the FDR set at 5%. Transcription factor-binding sites on collections of transcripts were identified with the HOMER (v4.6) software (Heinz *et al*, 2010) and its accompanying collection of tools. We limited the searches to 500-bp upstream of the transcription start sites. For most RNA-seq experiments, group sizes of *n* = 3 were chosen.

### Antibiotic-mediated depletion of intestinal bacteria

Mice were given an antibiotics cocktail of 100 mg/l ciprofloxacin (Sigma), 500 mg/l ampicillin (Sigma), 500 mg/l metronidazole (Sigma), and 250 mg/l vancomycin (Duchefa Labconsult) that was added to the drinking water for 3 weeks, as described in Van Hauwermeiren *et al* (2015). When antibiotics were given per oral 0.6 mg ciprofloxacin, 3 mg ampicillin, 3 mg metronidazole, and 1.5 mg vancomycin were gavaged daily in 100 μl drinking water. Depletion of the intestinal microflora was confirmed by culturing fecal samples on brain–heart infusion (BBL) plates in both aerobic and anaerobic conditions.

**Table 1.** Overview of qPCR primers and sequences

| Gene | Forward primer (5′-3′) | Reverse primer (5′-3′) |
| --- | --- | --- |
| h36B4 | CATGCTCAACATCTCCCCCTTCTCC | GGGAAGGTGTAATCCGTCTCCACAG |
| hCyclophilin A | TCCTGGCATCTTGTCCATG | CCATCCAACCACTCAGTCTTG |
| hTSC22D3 | GGAGATCCTGAAGGAGCAGA | TTCAGGGCTCAGACAGGACT |
| hDUSP1 | ACCACCACCGTGTTCAACTTC | TGGGAGAGGTCGTAATGGGG |
| hFKBP1 | GCCACATCTCTGCAGTCAAA | TCCCTCGAATGCAACTCTCT |
| hSGK1 | CCTCCACCAAGTCCTTCTCA | CCCTTTCCGATCACTTTCAA |
| mRpl | CCTGCTGCTCTCAAGGTT | TGGCTGTCACTGCCTGGTACTT |
| mGapdh | TGAAGCAGGCATCTGAGGG | CGAAGGTGGAAGAGTGGGAG |
| mHprt | AGTGTTGGATACAGGCCAGAC | CGTGATTCAAATCCCTGAAGT |
| mVillin | TCAAAGGCTCTCTCAACATCAC | AGCAGTCACCATCGAAGAAGC |
| mNr3c1 | AGCTCCCCCTGGTAGAGAC | GGTGAAGACGCAGAAACCTTG |
| mAgtpbp1 | GGGGTCGAAGAGCGAGTTTC | GAATGGAGTGAGTCTGCACCA |
| mIl6 | GGTGAAGACGCAGAAACCTTG | AGTGTCCCAACATTCATATTGTCAG |
| mIl1β | CACCTCACAAGCAGAGCACAAG | GCATTAGAAACAGTCCAGCCCATAC |
| Cxcl9 | TCCTTTTGGGCATCATCTTCC | TTTGTAGTGGATCGTGCCTCG |
| mSlc30a2 | CGGAGCCCGGTCCTTCTTA | GCATGGCAATAATGGTTGCTCT |
| EUB (UNIF334 and UNIF514) | ACTCCTACGGGAGGCAGCAGT | ATTACCGCGGCTGCTGGC |
| mMlkl | AATTGTACTCTGGGAAATTGCCA | TCTCCAAGATTCCGTCCACAG |
| mZbp1 | AAGAGTCCCCTGCGATTATTTG | TCTGGATGGCGTTTGAATTGG |
| mRipk3 | GTGCTACCTACACAGCTTGGA | CCCTCCCTGAAACGTGGAC |
| mDefa6 | CCAGGCTGATCCTATCCAAA | GTCCCATTCATGCGTTCTCT |
| mLyz1 | ATGGAATGGCTGGCTACTATGG | ACCAGTATCGGCTATTGATCTGA |
| mStat1 | TCACAGTGGTTCGAGCTTCAG | GCAAACGAGACATCATAGGCA |
| mUbc | AGGTCAAACAGGAAGACAGACGTA | TCACACCCAAGAACAAGCACA |

## Ileum slurry toxicity

Mice were euthanized; the terminal 8 cm of ileum was prepared; and the contents were weighed, isolated in a petri dish, minced and mixed, and frozen at 100 mg/ml in a solution containing sterile PBS. Seven hundred and fifty microlitre of this ileum slurry was injected i.p. in germ-free C57BL/6J mice and lethality recorded.

## Tissue sections and immunohistochemistry staining

Histopathology and immunostaining of ileum sections. Tissues were fixed with PFA, embedded in paraffin, and sectioned at 4μm. For hematoxylin and eosin staining, sections were dewaxed and stained with hematoxylin (Fluka, Diegem, Belgium) and eosin (Merck, Leuven, Belgium). The degree of damage was evaluated on entire organ sections by three observers in a blinded manner. Intestinal damage is characterized by decreased villus height, epithelial cell death at the villus top and loss of mucus layer and goblet cells. Taking into account all histological features, a damage score ranging from 0 (normal) to 4 (abnormal) was given to each mouse. For immunostaining, sections were dewaxed and boiled in 10 mM sodium citrate buffer for antigen retrieval, incubated for 1 h in blocking buffer (10 mM Tris–HCl pH 7.4, 0.1 M MgCl$_2$, 0.5%

Tween-20, 1% BSA, and 5% serum) and incubated with anti-MMP7 (1/100 dilution; 3801; Cell Signaling Technology). Fluorescent images and light microscopy images were taken by a laser scanning confocal microscope (Leica TCS SP5) and an Olympus light microscope, respectively.

## Transmission electron microscopy

Ileum tissue was cut into small pieces and immersed in a fixative solution of 2.5% glutaraldehyde and 3% formaldehyde in Na-cacodylate buffer 0.1 M, placed in a vacuum oven for 30 min and left rotating for 3 h at room temperature. This solution was later replaced with fresh fixative, and samples were left rotating overnight at 4°C. After washing, samples were post-fixed in 1% OsO$_4$ with K$_3$Fe(CN)$_6$ in 0.1 M Na-cacodylate buffer, pH 7.2. After washing in ddH$_2$O, samples were subsequently dehydrated through a graded ethanol series, including a bulk staining with 2% uranyl acetate at the 50% ethanol step followed by embedding in Spurr's resin. To select the area of interest on the block and in order to have an overview of the phenotype, semi-thin sections were first cut at 0.5 μm and stained with toluidine blue. Ultrathin sections of a gold interference color were cut using an ultramicrotome (Leica EM UC6), followed by a post-staining in a Leica EM AC20 for 40 min in uranyl acetate at 20°C and for

10 min in lead stain at 20°C. Sections were collected on formvar-coated copper slot grids. Grids were viewed with a JEM 1400plus transmission electron microscope (JEOL, Tokyo, Japan) operating at 80 kV.

### Relative quantification of Paneth cell markers

As described recently (Van Looveren *et al*, 2020), death of Paneth cells was quantified by measuring the expression of typical Paneth cell genes, by qPCR, i.e., *Defa6* and *Lyz1*.

### Bacterial counts in spleens and MLNs

Eighteen hours after injection of mice with PBS or 45 µg TNF, mice were euthanized and the spleen and mesenteric lymph nodes were isolated and collected in 1 ml sterile PBS. By using the TissueLyzer, the spleen and MLNs were homogenized in 200 µl brain–heart infusion medium (Becton Dickinson, Erembodegem, Belgium), plated onto tryptic soy agar (TSA) plates, and incubated at 37°C. The following day, the number of colony-forming units per ml was determined.

### Intestinal permeability

An *in vivo* permeability assay was performed using FITC-dextran as described previously (Van Looveren *et al*, 2020). Three hours after TNF challenge, 200 µl FITC-dextran (25 mg/ml in PBS) was administered by oral gavage. Five hours later, blood was collected in an EDTA-coated tube and centrifuged at 1,018 $g$ for 20 min at 4°C. Plasma was collected, and fluorescence was measured ($\lambda$exc/$\lambda$em = 488/520 nm).

### 16S rRNA sequencing

Fecal DNA was prepared with the QIAamp Fast DNA Stool mini kit (Qiagen) according to the manufacturer's instructions. The DNA was used for 16S rRNA library preparation using the Illumina NextEra XT v2 protocol, and sequencing was done on an Illumina MiSeq (paired end reads (2 × 300 bp), 8 bp index) by the VIB Nucleomics Core (Leuven, Belgium). Raw read quality control and preprocessing was done with Trimmomatic (leftover index removal and low-quality pair removal). Data were processed with Qiime1 starting from demultiplexed Illumina reads and using the Silva database (for open-reference OTU picking).

### Statistical analysis

All data were analyzed with GraphPad prism, except for the sequencing data and the proteomics data as indicated before. Survival curves were compared with a chi-square test. Data are represented as mean ± SD. Statistical differences between groups were calculated by means of Student's *t*-test, 1-way ANOVA, or 2-way ANOVA, as indicated in the figure legends. Samples were assumed to be normally distributed with similar variance between groups. No randomization was used to determine experimental groups, and no blinding of the investigator was performed. Group sizes were determined on the basis of previous experience. No data were excluded from the analyses.

**The paper explained**

**Problem**

Zinc has therapeutic effects, mainly in intestinal infections and diarrhea. The mechanism of this zinc effect is poorly known. Zinc also protects against intestinal effects induced by the cytokine tumor necrosis factor (TNF), which plays a key role in Crohn's disease. Since zinc was previously shown to protect in the TNF model in mice, we applied this model to investigate zinc's mode of action.

**Results**

By applying RNA-seq in ileum biopsies in mice treated with water or ZnSO4 in the drinking water, we found that zinc causes strong down-regulation of a large group of genes, known as ISRE/IRF genes. These are thought to (confirmed here) be induced in intestinal epithelium by gut microbes. Zinc lowers these genes, by actively and directly modulating the composition of the gut microbes, and our data suggest that mainly *Staphylococci* would be targeted by zinc. The reduction in ISRE/IRF genes is important, because several of these genes play an important role in killing certain epithelial cells (known as Paneth cells), when TNF is present. Because of the effects of zinc on bacteria and ISRE/IRF genes in these cells, they resist killing by TNF. How this leads to survival of the animals is not yet clear, but since zinc also reduces the escape of bacteria from the gut to other organs, the impact of zinc on the gut microbes or on the Paneth cell's death appears responsible as well. Mice with extremely high ISRE/IRF gene expression (e.g., because they have no adrenals) or extremely low ISRE/IRF genes, because they have no gut microbes, are not susceptible to the beneficial effects of zinc.

**Impact**

Now that the mechanism of zinc is unfolded, more precise zinc targeting or zinc replacing strategies in intestinal diseases where TNF is implicated (Crohn's disease) can be considered.

## Data availability

RNA-seq data: Gene expression. Deposited at the National Center for Biotechnology Information Gene Expression Omnibus public database (http://www.ncbi.nlm.nih.gov/geo/) under accession numbers GSE156119 (zinc ileum (C57BL/6J)), GSE156242 (zinc GR$^{WT}$ or GR$^{Dim}$), and GSE156282 (zinc liver (C57BL/6J)).

**Expanded View** for this article is available online.

### Acknowledgements

We would like to thank Prof. Jan Tuckermann (Ulm, Germany) for providing GR$^{fl/fl}$ as well as GR$^{Dim}$ mice, Prof. Walter Schaffner (Zurich, Switzerland) for MTF1$^{fl/fl}$ mice and Prof. Lynn Matrisian (CA, USA) for MMP7$^{-/-}$ mice. The MALDI-TOF MS was financed by the Research Foundation Flanders (FWO-Vlaanderen) as Hercules project [G0H2516N, AUGE/15/05]. Joke Vanden Berghe, Sara Van Ryckeghem, and animal caretakers are thanked for excellent animal care. Research in the author's laboratory was funded by the Agency for Innovation of Science and Technology in Flanders (IWT), the Research Council of Ghent University (GOA Program), the Research Foundation Flanders (FWO-Vlaanderen), the FWO Hercules program and Flanders Institute for Biotechnology (VIB).

### Author contributions

J.S. performed and guided about a third of the work and wrote parts of the manuscript, S.T. performed all bio-informatics analysis of RNA sequencing and 16S rRNA

sequencing analysis. T.Va., K.V., and C.W. were involved in the antibiotic experiments, the sampling for 16S rRNA sequencing and qPCRs. E.G., S.V., F.B., and F.V.I. performed the MALDI-TOF experiments; L.D.C., R.T., and J.R. did 16S rRNA sequencing; T.Ve. and L.F.M. performed the germ-free experiments, T.H. generated Slc30a2$^{-/-}$ mice. M.E., L.A., S.D., M.B., J.V., and S.V. provided technical assistance, R.D.R. and M.D.B. performed the TEM experiments. K.D. and R.B. helped with scientific guidance. C.L. supervised the research and wrote the manuscript.

## Conflict of interest

The authors declare that they have no conflict of interest.

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
