## [Review Process File · EMBO Molecular Medicine]

Zinc inhibits lethal shock by preventing microbe-induced interferon signature in intestinal epithelium

Jolien Souffriau, Steven Timmermans, Tineke Vanderhaegen, Charlotte Wallaey, Kelly Van Looveren, Lindsay Aelbrecht, Sylviane Dewaele, Jolien Vandewalle, Evy Goossens, Serge Verbanck, Filip Boyen, Melanie Eggermont, Lindsey De Commer, Riet De Rycke, Michiel De Bruyne, Raul Tito, Marlies Ballegeer, Sofie Vandevyver, Tiago Velho, Luis Ferreira Moita, Tino Hocheplied, Karolien De Bosscher, Jeroen Raes, Filip Van Immerseel, Rudi Beyaert, and Claude Libert

DOI: [10.15252/emmm.201911917](https://doi.org/10.15252/emmm.201911917)

Corresponding author: [Claude Libert \(claude.libert@irc.vib-ugent.be\)](mailto:claude.libert@irc.vib-ugent.be)

Review Timeline:

Submission Date:	18th Dec 19
Editorial Decision:	24th Jan 20
Revision Received:	16th Jun 20
Editorial Decision:	9th Jul 20
Revision Received:	19th Aug 20
Accepted:	19th Aug 20

Editor: *Celine Carret*

Transaction Report:

24th Jan 2020

Dear Claude,

Thank you for the submission of your manuscript to EMBO Molecular Medicine. We have now heard back from the two referees whom we asked to evaluate your manuscript.

You will see that both referees find this work interesting, while still in need of additional supportive and more definitive data. Both referees suggest additional experiments to support causality in vivo and ref. #2 requests more mechanistic insights. Ref. #1 wants more supportive experiments as well strengthening of the conclusions. We would like to invite a major revision of your study and increase the deadline for resubmission to at least 9 months (not firm, if you need more, please do let us know) in order to be able to perform the non trivial requested experiments with germ-free isolator technology as we believe that these in vivo additional data would considerably improve the significance and conclusiveness of your work. Adding mechanism would be desirable, but at this stage not mandatory.

We would therefore welcome the submission of a revised version within three months for further consideration and would like to encourage you to address all the criticisms raised as suggested to improve conclusiveness and clarity. Please note that EMBO Molecular Medicine strongly supports a single round of revision and that, as acceptance or rejection of the manuscript will depend on another round of review, your responses should be as complete as possible.

Please also contact us as soon as possible if similar work is published elsewhere. If other work is published we may not be able to extend the revision period beyond nine months.

I look forward to receiving your revised manuscript.

Yours sincerely,

Celine Carret

Celine Carret, PhD
Senior Editor
EMBO Molecular Medicine

**** Reviewer's comments ****

Referee #1 (Comments on Novelty/Model System for Author):

The authors have studied appropriate mutant mouse models on GR signaling, antibiotic treatment protocols and germ-free mouse models, which strengthen their claims. The manuscript provides new information on the influence of the gut microbiota on glucocorticoid signaling, which could be translated to clinically relevant situations.

Referee #1 (Remarks for Author):

Souffriau and colleagues provide an interesting and innovative manuscript on the role of the micronutrient zinc and its protective impact on the TNF- α -induced systemic inflammatory response syndrome, which depends on glucocorticoid signaling through the activation of interferon-stimulated response genes, largely influenced by the presence of gut microbiota. The study describes a zinc-dependent modulation of the gut microbiota, which the authors linked to reduced induction of ISRE/IRF genes in the gut epithelium. Souffriau et al. present interesting work that is based on survival analyses of GR mutant mice, pharmacological interventions, RNA-sequencing-based interaction analyses and qRT-PCRs, histology and taxonomic microbiome analyses. While the overall theme and the experiments of this study are of extraordinary high biomedical interest and the experimentation could principally fit with the scope of EMBO Molecular Medicine, a number of experiments are missing that are required to prove causality and to support the claims of authors. In particular the epithel-specificity of the identified signaling cue, the link to upstream epithelial TLR signaling and the causal link of this pathway to the microbiota require additional experimentation. I have added some comments, which the authors should address when revising their manuscript.

Major comments:

1. How do zinc plasma levels (25 mM administered via the drinking water) relate to average zinc plasma levels of conventionally-raised vs germ-free mice? Is the uptake of this micronutrient influenced by the presence of the gut microbiota and are the plasma levels achieved by administration via the drinking water still in the physiologic range? Remarkably, zinc blood levels are reduced in the ZnSO₄-supplemented GR^{Dim} mice relative to GR^{WT} (Fig. 1E). Why was zinc-flux altered in the mutant mice?
2. To address the zinc-mediated effect of corticosteroids on the epithelial transcriptome of the ileum, it would significantly improve the study to include RNA-sequencing of isolated epithelial cells or at least from ileal tissues from the generated tissue specific Villin-Cre x GR^{fl/fl} mouse model. Here it would be interesting if the identified up- and down-regulated genes interact with the genes identified in the Dex-treatment model shown in Fig. 2A, B. Also the overlay with the identified genes in whole ileum would be interesting. In that way, the epithel-specific influence is addressed.
3. The fecal transplant experiment with cecal content from ZnSO₄-treated mice and untreated controls needs to be performed under germ-free isolator conditions by colonization of germ-free C57BL/6 mice and lethality should be recorded. Also it should be shown taxonomic analyses that the transplanted microbiota is similar to the donor microbiota.

4. To rigorously test the influence of the microbiota on ISRE/IRF gene expression via epithelial GR signaling, as suggested by the authors, the generated Villin-Cre x GRfl/fl mouse model should be rederived as germ-free and the supplementation of zinc should be compared with this elegant genetic model along with conventionally-reared controls. Although the authors provide results pointing to a microbiota/GR-mediated regulation of ISRE/IRF gene expression by zinc, this experiment is necessary to prove this pathway. If the germ-free model is not available, at least the antibiotic treatment protocol should be performed on the Villin-Cre x GRfl/fl mouse model. To show the effect on ISRE/IRF gene expression in antibiotics-treated WT mice is not sufficient.
5. The upstream signaling mechanism was suggested to be mediated through epithelial TLR4. For clarity, results on the exact ISRE/IRF gene expression markers shown in Fig. 3 should also be included in this main figure for the Tlr4-flox x Vil-Cre mouse model. Also additional Tlr-flox mouse models (e.g. TLR2 or TLR5) should be tested to pinpoint the specificity of the proposed pathway.
6. The Paneth cell analyses on the MMP-7 marker in Fig. 4A are not convincing. Here a much better resolution is needed and a specificity control (e.g. MMP-7 knock-out) would add.
7. How does zinc prevent the influx of bacteria from the gut lumen that was nicely demonstrated in Fig. 4B? Is the assembly of epithelial tight junctions improved? Is the zinc-dependent protection only localized to the crypts?
8. Referring to Fig. 5C, in an attempt to prove causality, it would strengthen the manuscript if at least one monocolonization experiment on germ-free mice with one of the zinc-dependently regulated microbial taxa would be performed with the TNF- α -induced systemic inflammatory response syndrome mouse model.
9. Overall, the description of the clinical relevance of the study should be improved and pointed out in the introduction.

Minor comments:

1. In paragraph 2 of the introduction, the authors should point out the clinical settings of dexamethasone administration to explain the translational aspect of their study.
2. The statement in the second sentence of the discussion, that TNF overexpression leads to inflammatory bowel disease and arthritis seems a bit blunt with respect to the myriad of TNF-functions. This part of the introduction needs refinement.
3. In Fig. 1G-I the authors should also control versus Villin-Cre x GRWt/WT mice to exclude an effect of Cre-recombinase in the experiment.
4. How was the average mouse weight at the beginning of antibiotic treatment protocol and how at the endpoint? Did the mice consume the same amount of drinking water containing 25 mM

ZnSO₄? In other words, can the lack of protection of zinc administration in the antibiotics-treated group be caused by dehydration or side-effects of the antibiotic treatment protocol? How was water consumption in the ZnSO₄-supplemented germ-free group relative to the conventionally-raised control group?

5. The figure presentation needs to be improved throughout. It would be good to show the individual data points in addition to the mean value in the bar graphs.

6. The number of individual mice of each group analysed in the RNA-sequencing analyses in Fig. 2A and B is very low (N=3). This limitation of the study needs to be discussed in the results description.

Referee #2 (Remarks for Author):

This manuscript provides a novel mechanistic insight for the protective role of Zinc in the TNF-induced model of systemic inflammatory response syndrome (SIRS). The authors claim that Zinc modulates the microbiota in a way that dampens the expression of ISRE/IRF genes by intestinal epithelial cells (IECs) and makes the mice more resistant to TNF-induced lethality, caused by intestinal epithelial cell (IEC) necroptosis and systemic bacterial dissemination. Interestingly, the authors show that combinatorial administration of Dexamethasone (DEX) and ZnSO₄ remarkably increases the survival of mice challenged with a lethal dose of TNF.

Major concerns

1/ The authors suggest that Zinc protects from TNF induced SIRS by shaping the gut microbiota. It is clearly shown that Zinc induces dysbiosis (Figure 4) as previously reported (Zackular et al, Nature Medicine, 2016) but a direct role of the Zinc-modulated microbiota in the TNF-induced model is poorly supported. The data presented in Figure 2G shows that the transfer of microbiota from mice pretreated with ZnSO₄, increase the survival in the cecal slurry injection model. However, the authors may consider performing a similar, more direct, microbiota transfer experiment in the TNF-induced SIRS model.

2/ The study shows that DEX and Zinc act via independent pathways to suppress ISRE/IRF gene expression by IECs. The surprising total rescue of mice treated with the combination of Dex with ZnSO₄ indicate additive but does not exclude independent protective pathways. To further support the claim that both pathways (Zinc and Dex) protect via suppression of STAT1 and ISRE/IRF expression authors should consider:

a/ Analyzing the STAT1 and ISRE/IRF expression in mice treated with combinatorial (Dex and ZnSO₄) and single treatments in the TNF induced SIRS model (Figure 5D).

b/ Including STAT1 KO mice in figure 5D, challenged with the same TNF dose (50 µg TNF/20 g bodyweight)

Minor concerns

- 1/ The authors may consider discussing that Zn can have a direct role in repressing IRF-related genes by binding to the newly identified Zinc finger IRF composite elements. (Ochiai et. al, Blood Advances, 2018)
- 2/ In Figure 1 (G-H) it is suggested that the partial protective effect of ZnSO₄ in GRVillKO mice is due to the specificity of the cre line. Are the authors considering that GR signaling in another cell type, such as macrophages could be equally important for the protection against TNF-induced lethality and could potentially have a role in the resistance of GRDim and Adx mice to ZnSO₄ treatment?
- 3/ Contradictory results are represented in Fig.2D-E. While antibiotic treated mice were partially rescued from TNF-induced lethality (Fig.2D), germ free mice were not protected (Fig.2F).
- 4/ In the RNA-seq results presented in figure 2A, gene set enrichment analysis should be performed in the common 48 genes of Figure 3B, to show common enriched pathways downstream of Zinc and GR signaling.
- 5/ In figure 4A, bigger magnification images are needed in H&E staining. Furthermore, in mice treated with ZnSO₄ and injected with TNF, apart from the degranulation of Paneth cells a more complete characterization of their phenotype is needed. The authors should assess the Paneth/Goblet cell number, the expression of antimicrobial peptides, the intestinal epithelial cell permeability, as well as the epithelial cell death (apoptosis and necroptosis).
- 6/ The resistance of GRDim, GRVillKO and Adx mice in ZnSO₄ treatment is attributed to the threshold of ISRE/IRF dependent gene expression. It might be worthwhile increasing the ZnSO₄ dosage in these mice to exclude a potential protective effect in higher doses.
- 7/ The authors should revise their reference list including publications showing that Zinc can modulates the gut microbiota composition. (Zackular et al, Nature Medicine, 2016 & Dalin Li et al, Gastroenterology 2016).
- 8/ Although there is a lot of literature on the anti-inflammatory role of Zinc, the authors report an independent function during TNF-induced SIRS. The data presented in SI Appendix, Figure S3, show that ZnSO₄ treatment although doesn't affect proinflammatory cytokines levels in the ileum, reduces IL-6 and Eotaxin serum levels. These results should be discussed further as IL-6 and Eotaxin are the major cytokines driving mortality in this model. In the same context, serum IL1 levels should also be measured.
- 9/ In figure 5A statistical significance in the microbiota composition between wild type and ZNSO₄ treated mice should be indicated.
- 10/ The working hypothesis as presented in figure 3I seems to be redundant as it is summarized in the graphical proposed mechanism (figure 5E).

***** Reviewer's comments *****

Referee #1 (Comments on Novelty/Model System for Author):

The authors have studied appropriate mutant mouse models on GR signaling, antibiotic treatment protocols and germ-free mouse models, which strengthen their claims. The manuscript provides new information on the influence of the gut microbiota on glucocorticoid signaling, which could be translated to clinically relevant situations.

Referee #1 (Remarks for Author):

Souffriau and colleagues provide an interesting and innovative manuscript on the role of the micronutrient zinc and its protective impact on the TNF- α -induced systemic inflammatory response syndrome, which depends on glucocorticoid signaling through the activation of interferon-stimulated response genes, largely influenced by the presence of gut microbiota. The study describes a zinc-dependent modulation of the gut microbiota, which the authors linked to reduced induction of ISRE/IRF genes in the gut epithelium. Souffriau et al. present interesting work that is based on survival analyses of GR mutant mice, pharmacological interventions, RNA-sequencing-based interaction analyses and qRT-PCRs, histology and taxonomic microbiome analyses. While the overall theme and the experiments of this study are of extraordinary high biomedical interest and the experimentation could principally fit with the scope of EMBO Molecular Medicine, a number of experiments are missing that are required to prove causality and to support the claims of authors. In particular the epithel-specificity of the identified signaling cue, the link to upstream epithelial TLR signaling and the causal link of this pathway to the microbiota require additional experimentation. I have added some comments, which the authors should address when revising their manuscript.

Major comments:

1. How are do zinc plasma levels (25 mM administered via the drinking water) relate to average zinc plasma levels of conventionally-raised vs germ-free mice? Is the uptake of this micronutrient influenced by the presence of the gut microbiota and are the plasma levels achieved by administration via the drinking water still in the physiologic range? Remarkably, zinc blood levels are reduced in the ZnSO₄-supplemented GRDim mice relative to GRWT (Fig. 1E). Why was zinc-flux altered in the mutant mice?

Good questions.

In the mice, the blood Zn levels are usually about 100 ug/dl, which is indeed according to the known physiological levels, which are known to fluctuate, in mammals, between 75 ug/dl and 125 ug/dl. In our hands, the levels increase after a one-week ZnSO₄ treatment to 200-300 ug/dl, depending on the experiment, which indeed is well above the physiological levels.

*We have added this information in the **results** section of the revised paper.*

*We have performed a new experiment in C57BL/6J mice bred and treated with normal drinking water or 7 days with 25 mM ZnSO₄ in a normal animal house, and C57BL/6J mice bred and treated similarly in a GF facility. The blood Zn levels were measured and found to increase in both groups of mice to a very similar level. The data are shown here as well as in the paper. So, in mice, we found that neither antibiotics, nor germ-free conditions changed the Zn uptake into the blood. These data, and the conclusion are added to the revised paper in figure 3, **results section and discussion**.*

As for the differences in blood Zn levels between GR^{WT} and GR^{Dim}, these were mainly due to a high degree of variation in the levels in individual mice. We have performed new experiments with GR^{WT} and GR^{Dim} mice with H₂O (n=10) and ZnSO₄ (n=6) and measured blood Zn levels again. The results show that there are no differences in response in both mouse strains, and we have now replaced the figure with this improved one.

2. To address the zinc-mediated effect of corticosteroids on the epithelial transcriptome of the ileum, it would significantly improve the study to include RNA-sequencing of isolated epithelial cells or at least from ileal tissues from the generated tissue specific Villin-Cre x GR^{f1/f1} mouse model. Here it would be interesting if the identified up- and down-regulated genes interact with the genes identified in the Dex-treatment model shown in Fig. 2A, B. Also the overlay with the identified genes in whole ileum would be interesting. In that way, the epithel-specific influence is addressed.

I would like to thank the reviewer as this is an interesting question indeed.

*In the paper, we state that Zn-induced and -reduced gene expressions in ileum epithelium are independent of GR activation. To make this statement harder than we did (by comparing Zn-induced with DEX-induced and Zn-repressed with Dex-repressed gene signatures) we have now performed new RNAseq experiments to study the Zn-induced and Zn-repressed signatures in GR^{WT} and in GR^{Dim} mice. We have chosen these mutants above GR^{villKO} because the latter show quite some shortcomings in terms of GR deficiency in the intestinal epithelium as we observe (by IHC) quite some villi that have no reduced GR levels because of imperfect cre penetrance, a well-known problem with villin-cre. GR^{Dim} mice have a point mutation in GR, making them hardly able to regulate genes. We thus have compared the induction of the 116 genes of Figure 2A in GR^{WT} and GR^{Dim} mice and we found that the majority of these genes (n=67) is also significantly induced in GR^{Dim} mice, while the other 49 genes are also induced but do not reach significance. By chi² analysis with Yates correction, or Fisher exact test, this high degree of resemblance of Zn induction patten in GR^{WT} and GR^{Dim} has p<0.0001. Also, plotting the Zn-induction levels observed by RNAseq in GR^{Dim} and GR^{WT}, and detection of regression curve, we observe a slope of the regression curve of 0.9833, suggesting a near perfect correlation of both datasets, and a Pearson correlation coefficient of 0.68, which is considered as a strong correlation. These data suggest that indeed Zn-upregulated genes are equally strong induced whether GR is active or not. Very similar data are obtained in case of Zn-repressed genes. Because these data have an added value for our paper, we have added them in the results section and in **Figure 2**.*

3. The fecal transplant experiment with cecal content from ZnSO₄-treated mice and untreated controls needs to be performed under germ-free isolator conditions by

colonization of germ-free C57BL/6 mice and lethality should be recorded. Also it should be shown taxonomic analyses that the transplanted microbiota is similar to the donor microbiota.

*This is an outstanding suggestion. We have performed a new experiment and produced new material from C57BL/6J mice treated with H₂O or with ZnSO₄ and injected in germ-free C57BL/6J mice, as suggested by the reviewer. We observed, basically, that the bacterial slurry derived from the ZnSO₄ treated mice was much less toxic when injected in the germ-free mice. These results add to the understanding of our findings and are added in **Figure 2** of the revised paper.*

4. To rigorously test the influence of the microbiota on ISRE/IRF gene expression via epithelial GR signaling, as suggested by the authors, the generated Villin-Cre x GR^{fl/fl} mouse model should be rederived as germ-free and the supplementation of zinc should be compared with this elegant genetic model along with conventionally-reared controls. Although the authors provide results pointing to a microbiota/GR-mediated regulation of ISRE/IRF gene expression by zinc, this experiment is necessary to prove this pathway. If the germ-free model is not available, at least the antibiotic treatment protocol should be performed on the Villin-Cre x GR^{fl/fl} mouse model. To show the effect on ISRE/IRF gene expression in antibiotics-treated WT mice is not sufficient.

We have taken this suggestion of the reviewer at heart. We have not been able to transfer GR^{VillKO} mice to a germ-free facility for three reasons, namely (1) because of the long waiting list (>1.5y) for embryo transfers (which were even completely stopped because of the Covid-19 lockdown until further notice), (2) because of the budget restriction (5.000E for transferring a mutant line into the GF facility). We have therefore decided to do the experiment with antibiotics. As in the paper, mice were treated for 2W with normal drinking water or with the AB cocktail. The third week, they were treated by gavage with AB or PBS and received water or ZnSO₄ in the drinking water (because the mice refuse to drink AB plus ZnSO₄). This was done in GR^{fl/fl} mice and in GR^{VillKO} mice. At the end of the third week the mice were killed and ileum mRNA was prepared and qPCR for ISGs and STAT1 were performed. All groups consisted of n=4. The experiment thus consisted in total of 32 mice. Unfortunately, the experiment failed twice, and now our GR^{VillKO} colony is exhausted and we have no means to apply for an ethical approval to do the experiment a third time. To our great sorry, this is the only experiment of the entire revision that we have not been able to perform. We trust that editor and reviewers, however, are convinced of the solidity of the study and paper in their entirety and can live with the absence of this control.

5. The upstream signaling mechanism was suggested to be mediated through epithelial TLR4. For clarity, results on the exact ISRE/IRF gene expression markers shown in Fig. 3 should also be included in this main figure for the Tlr4-flox x Vil-Cre mouse model. Also additional Tlr-flox mouse models (e.g. TLR2 or TLR5) should be tested to pinpoint the specificity of the proposed pathway.

As suggested by the reviewer, we have now measured the expression of key ISRE/IRF genes of our study, in ileum samples of additional TLRKO mouse models, by qPCR and have incorporated

these data in **the results section** of the revised paper. The expressions of the ISRE genes *Ripk3*, *Mlkl* and *Zbp1* were measured in TLR2^{-/-}, TLR4^{-/-}, TLR5^{-/-} and TLR9^{-/-} mice. The description of the data is included in the results section and discussion. As we already suggested in the first version of the paper, TLR4 seems to play a role in the establishment of the ISRE gene expression in ileum, but the impact of loss of TLR4 suggest that more factors play a role. The data also reveal that TLR2 may play a rather repressive role, and that the loss of TLR5 and TLR9 have only limited impact. Since the number of candidate-mediating molecules is high (*cGAS*, *STING*, *MyD88*, *TRIF*, *NLRs*, other TLRs) and since we have no time nor means to investigate the adaptations of bacterial compositions in these TLR-deficient mice, we believe that a detailed investigation of this aspect has to be performed at a later stage, for example once we have found ways to isolate Paneth cells and study them in high throughput, in our further projects.

6. The Paneth cell analyses on the MMP-7 marker in Fig. 4A are not convincing. Here a much better resolution is needed and a specificity control (e.g. MMP-7 knock-out) would add.

We agree with the reviewer.

We have done the following to ameliorate this part of the paper.

- 1. We have thawed MMP7^{-/-} embryo's, transferred them to fosters and at the age of 8W took an ileum sample of these mice as well as MMP7^{+/+} mice, to stain for MMP7 with the antibody and found indeed no staining in the former mice and nice staining in the latter. This control was added to the figure as a control. These data are mentioned in the revised paper in the results section and the stainings shown in **Supplemental Figures**.*
- 2. We have also performed Experiments in normal mice to study the effects of PBS or TNF injection in mice that had been treated for a week with H₂O or ZnSO₄ and made tissue sections for Transmission Electron Microscopy (TEM). The pictures are very revealing. They show nicely the devastating cell-death effects of TNF on Paneth cells (extrusion in the lumen and loss of contact with the basal membrane) and that Zn is able to protect. These data are added in the revised paper in the **results** section.*
- 3. We have also performed studies with mice treated as mentioned in point 2 (10 mice per group, total numbers n=40) and taken ileum samples and done qPCR to detect Paneth cell specific markers qPCR, the decline of which being a readout of less PCs. These data confirm that PC numbers decline by TNF, but significantly less by Zn treatment. These data are added in the revised paper in the **results** section.*

7. How does zinc prevent the influx of bacteria from the gut lumen that was nicely demonstrated in Fig. 4B? Is the assembly of epithelial tight junctions improved? Is the zinc-dependent protection only localized to the crypts?

These are questions that have kept us busy while developing the project and the paper. The major aim of our paper is to show our findings, which strongly suggest that Zn has a direct impact on the composition of the microbiota. Since the microbiota strongly determines the

expression of ISRE/IRF genes in the ileum (very likely the Paneth cells as suggested by other papers), and since some ISRE/IRF genes are mediators of necroptosis, the decrease in ISRE/IRF gene expression by Zn (via microbiota) makes these cells less susceptible to TNF-induced necroptosis. This is shown in Figure 5 and 6: we suggest that this cell death of Paneth cells leads to local loss of tissue integrity, and that this is the route via which bacteria find their way to the periphery. This last is something we have not been able to prove or disprove and that we are working on very hard. This sequence of events (with emphasis of the facts and hypothesis) has been described clearer in the discussion of the revised paper.

8. Referring to Fig. 5C, in an attempt to prove causality, it would strengthen the manuscript if at least one monocolonization experiment on germ-free mice with one of the zinc-dependently regulated microbial taxa would be performed with the TNF- α -induced systemic inflammatory response syndrome mouse model.

*This is an obvious question, which we have been asking ourselves in the past, and have tried to answer by transplantation and co-housing studies in the past, using germ-free mice and antibiotics treated mice, but without success. For the revision of the paper, we have switched gears and considered another option taking four datasets in consideration: (1) our data repeatedly have led us to the ileum as the site of action of zinc (zinc protects against TNF-induced Paneth cell death), (2) we described in the first version of the paper, that TNF led to evasion from the lumen of several bacteria into spleen and mesenteric lymph nodes and that zinc leads to strong reduction of this aspect, mainly reducing the amounts of Staphylococci in these tissues (3) as zinc has strong effects (on the genus level) on fecal microbiota and (4) since the toxicity of injected ileum contents from zinc-treated mice is less than from water treated mice in GF mice, we decided on the following strategy. First, we wanted to study the differences in ileum composition and, since we were interested in aerobic bacteria and species level, we decided to perform similar experiments as we did with the MLNs and spleen contaminations. We treated mice with water (n=5) and with zinc (n=5) for a week, then isolated ileum slurry, pooled, normalized the weight, plated out on TSA plates, counted the colonies after 24h and randomly picked >100 clones from the water (n=110) and zinc (n=102) conditions, and identified all of these by MALDI-TOF. The data revealed that in normal (water) ileum the abundant species of Staphylococcus sciuri (61% of colonies) and Staphylococcus nepalensis (21%) are entirely gone after zinc treatment for a week. **These new data are added to the manuscript.** Based on our in vitro studies, using zinc-containing TSA plates, this effect of zinc is likely to be a direct bactericidal effect.*

*Based on the hypothesis of the paper (that ileum bacteria induce an ISG response in Paneth cells, sensitizing them to TNF-induced cell death), we considered that the most informative transplantation experiment should be to study if mice, protected against TNF by zinc treatment can be re-sensitized to TNF by transplanting them with Staphylococcus. This is what we have done, with success. Mice were treated with H₂O or ZnSO₄ for 7 days, after which the ZnSO₄ was removed and replaced by H₂O. 12h and 24h later, the mice were transplanted with 10⁸ S. sciuri or 10⁸ S. nepalensis, and 3h later TNF was injected. The data, incorporated in the **results section** of the revised paper, illustrate that Zn-induces protection against TNF lethal shock, but that this Zn protection is entirely reverted by transplanting mice with these bacteria.*

9. Overall, the description of the clinical relevance of the study should be improved and pointed out in the introduction.

We have done our best to improve this in the paper's introduction and discussion, as suggested.

Minor comments:

1. In paragraph 2 of the introduction, the authors should point out the clinical settings of dexamethasone administration to explain the translational aspect of their study.

We have added this to the introduction as suggested.

2. The statement in the second sentence of the discussion, that TNF overexpression leads to inflammatory bowel disease and arthritis seems a bit blunt with respect to the myriad of TNF-functions. This part of the introduction needs refinement.

We have changed this part of the paper as suggested.

3. In Fig. 1G-I the authors should also control versus Villin-Cre x GRWt/WT mice to exclude an effect of Cre-recombinase in the experiment.

With all due respect for the reviewer, but we have not been able to address this remark, because, we had no ethical clearance for this experiment and had no mouse space & cages for it in our allowed slots, this type of control experiment had to be dropped. Although we, as scientists, just like the reviewer, are quite fanatic when it comes to controls, there is no real reason to believe that cre would have any impact on our phenotypes, either zinc response or TNF response.

4. How was the average mouse weight at the beginning of antibiotic treatment protocol and how at the endpoint? Did the mice consume the same amount of drinking water containing 25 mM ZnSO₄? In other words, can the lack of protection of zinc administration in the antibiotics-treated group be caused by dehydration or side-effects of the antibiotic treatment protocol? How was water consumption in the ZnSO₄-supplemented germ-free group relative to the conventionally-raised control group?

Point well taken. We have never seen any obvious signs of drop in body weight by the treatments. The new experiments we have performed to reply to major comment 4 have been used to measure the body weights of the mice when put on normal drinking water and on antibiotic and/or ZnSO₄ containing drinking water and have recorded body weights in function of time. As can be seen in the figure, there is no sign of dehydration. We are

showing these data for the interest of the reviewer, but are not showing them in the revised paper. We do mention however, in the results section, that there is no sign of weight loss or dehydration in mice that are not protected.

5. The figure presentation needs to be improved throughout. It would be good to show the individual data points in addition to the mean value in the bar graphs.

We have done our best to improve the figures, show the individual data points and other improvements.

6. The number of individual mice of each group analyzed in the RNA-sequencing analyses in Fig. 2A and B is very low (N=3). This limitation of the study needs to be discussed in the results description.

We have addressed this concern, as suggested, in the revised paper.

Referee #2 (Remarks for Author):

This manuscript provides a novel mechanistic insight for the protective role of Zinc in the TNF-induced model of systemic inflammatory response syndrome (SIRS). The authors claim that Zinc modulates the microbiota in a way that dampens the expression of ISRE/IRF genes by intestinal epithelial cells (IECs) and makes the mice more resistant to TNF-induced lethality, caused by intestinal epithelial cell (IEC) necroptosis and systemic bacterial dissemination. Interestingly, the authors show that combinatorial administration of Dexamethasone (DEX) and ZnSO₄ remarkably increases the survival of mice challenged with a lethal dose of TNF.

Major concerns

1/ The authors suggest that Zinc protects from TNF induced SIRS by shaping the gut microbiota. It is clearly shown that Zinc induces dysbiosis (Figure 4) as previously reported (Zackular et al, Nature Medicine, 2016) but a direct role of the Zinc-modulated microbiota in the TNF-induced model is poorly supported. The data presented in Figure 2G shows that the transfer of microbiota from mice pretreated with ZnSO₄, increase the survival in the cecal slurry injection model. However, the authors may consider performing a similar, more direct, microbiota transfer experiment in the TNF-induced SIRS model.

*This is a very relevant question, which we have been asking ourselves in the past, and have tried to answer by transplantation and co-housing studies in the past, using germ-free mice and antibiotics treated mice, but without success. For the revision of the paper, we have switched gears and considered another option taking four datasets in consideration: (1) our data repeatedly have led us to the ileum as the site of action of zinc (zinc protects against TNF-induced Paneth cell death), (2) we described in the first version of the paper, that TNF led to evasion from the lumen of several bacteria into spleen and mesenteric lymph nodes and that zinc leads to strong reduction of this aspect, mainly reducing the amounts of Staphylococci in these tissues (3) as zinc has strong effects (on the genus level) on fecal microbiota and (4) since the toxicity of injected ileum contents from zinc-treated mice is less than from water treated mice in GF mice, we decided on the following strategy. First, we wanted to study the differences in ileum composition and, since we were interested in aerobic bacteria and species level, we decided to perform similar experiments as we did with the MLNs and spleen contaminations. We treated mice with water (n=5) and with zinc (n=5) for a week, then isolated ileum slurry, pooled, normalized the weight, plated out on TSA plates, counted the colonies after 24h and randomly picked >100 clones from the water (n=110) and zinc (n=102) conditions, and identified all of these by MALDI-TOF. The data revealed that in normal (water) ileum the abundant species of Staphylococcus sciuri (61% of colonies) and Staphylococcus nepalensis (21%) are entirely gone after zinc treatment for a week. **These new data are added to the manuscript.** Based on our in vitro studies, using zinc-containing TSA plates, this effect of zinc is likely to be a direct bactericidal effect.*

Based on the hypothesis of the paper (that ileum bacteria induce an ISG response in Paneth cells, sensitizing them to TNF-induced cell death), we considered that the most informative transplantation experiment should be to study if mice, protected against TNF by zinc treatment can be re-sensitized to TNF by providing them with Staphylococcus. This is what we have done,

with success. Mice were treated with H₂O or ZnSO₄ for 7 days, after which the ZnSO₄ was removed and replaced by H₂O. 12h and 24h later, the mice were transplanted with 10⁸ *S. sciuri* or 10⁸ *S. nepalensis*, and 3h later TNF was injected. The data, incorporated in **the results section** of the revised paper, illustrate that Zn induces protection against TNF lethal shock, but that this Zn protection is entirely reverted by transplanting mice with these bacteria.

2/ The study shows that DEX and Zinc act via independent pathways to suppress ISRE/IRF gene expression by IECs. The surprising total rescue of mice treated with the combination of Dex with ZnSO₄ indicate additive but does not exclude independent protective pathways. To further support the claim that both pathways (Zinc and Dex) protect via suppression of STAT1 and ISRE/IRF expression authors should consider:

a/ Analyzing the STAT1 and ISRE/IRF expression in mice treated with combinatorial (Dex and ZnSO₄) and single treatments in the TNF induced SIRS model (Figure 5D).

Very good remark. We have performed new experiments to study ISG genes and Stat1 in the ileum via qPCR in mice pretreated (a) with normal drinking H₂O (1 week) and then injected with PBS or DEX and then challenged with PBS or TNF or (b) with 25 mM ZnSO₄, PBS/DEX and PBS/TNF. 6h after challenge with PBS/TNF ileum samples were taken.

*The results are in line with the expectations, based on the hypothesis. The expression data of two critical ISG genes in our work, *Mi1* and *Zbp1*, are induced by TNF, and this TNF effect is nicely repressed by DEX. These effects are observed in H₂O or ZnSO₄ pretreated groups, but significantly more outspoken in the latter's groups compared to the former's groups. As a result, TNF-induced ISG gene induction is significantly lower in Zn/DEX treated mice than in H₂O/DEX treated mice. The additive effect of DEX and Zn is also observed when studying the *Stat1* expression levels. These interesting data have been introduced in the **results section** of the revised paper.*

b/ Including STAT1 KO mice in figure 5D, challenged with the same TNF dose (50 µg TNF/20 g bodyweight)

*We agree with the author that response of STAT1^{-/-} mice to TNF would be an added value for the paper. In an earlier paper, (Ballegeer, J. Clin. Invest., 2019), we described that these mice are remarkably resistant to TNF. We have now performed new experiments, using the exact condition of TNF dose as used in the old Figure 5 (now Figure 7), i.e. 50 µg, and challenged STAT1^{-/-} and STAT1^{+/+} littermates and found a robust protection, which is now added to **the results** section of the revised paper.*

Minor concerns

1/ The authors may consider discussing that Zn can have a direct role in repressing IRF-related genes by binding to the newly identified Zinc finger IRF composite elements. (Ochiai et. al, Blood Advances, 2018).

This is a great remark and we have integrated this in the discussion of the paper.

2/ In Figure 1 (G-H) it is suggested that the partial protective effect of ZnSO₄ in GRVillKO mice is due to the specificity of the cre line. Are the authors considering that GR signaling in another cell type, such as macrophages could be equally important for the protection against TNF-induced lethality and could potentially have a role in the resistance of GRDim and Adx mice to ZnSO₄ treatment?

*Very good point. It would certainly be an interesting alternative to our explanation. We have discussed this point in the **Discussion**. We have considered to start crosses with GR^{fl/fl} and other tissue-specific cre mice, but after all we have thought it more wise to suggest this alternative explanation in the paper, and perform these lengthy and uncertain, costly experiments at a later stage for follow-up work, as they are, basically not going to change the basis of the paper, i.e. the focus on the role of Zn on the triangle microbiota-Paneth cells-GR.*

3/ Contradictory results are represented in Fig.2D-E. While antibiotic treated mice were partially rescued from TNF-induced lethality (Fig.2D), germ free mice were not protected (Fig.2F).

*The reviewer hits the spot. We have observed in many experiments that antibiotics protect against TNF (see also Van Hauwermeiren et al., Mucosal Immunology 2015 and Van hauwermeiren et al., J. Clin. Invest., 2013). The most obvious explanation deals with the role of mucosal bacteria in the TNF-induced lethality, which is a key aspect in our studies. One would indeed expect GF mice to be protected to TNF as well. As can be seen in Fig. 2F, the GF mice are actually dying quicker than the normal mice from the same TNF dose. We have got permission from the ethics committee to run a couple of extra experiments. We have injected TNF (intravenously, to make sure that we do not inject into the huge cecum in GF mice while injecting intraperitoneally), normal C57BL/6J mice with 8 ug TNF, 10 ug TNF or 20 ug and found 8 ug to be a useful dose. We the injected C57BL/6J GF mice and control C57BL/6J mice and found that GF mice 5 deaths/7 while in control mice 5 deaths/9. These data confirm that there is no obvious protection in GF mice. The contradiction with antibiotics has been observed in other model systems and may be the result of developmental shortcomings of GF mice: they have a lack of mucosal cell regeneration, digestive activity, mucosa-associated lymphoid tissue, lamina propria cellularity, muscle layer thickness, resistance to certain infections etc. We have decided to point the readers to this this contradiction in the **results and discussion** of the revised paper.*

4/ In the RNA-seq results presented in figure 2A, gene set enrichment analysis should be performed in the common 48 genes of Figure 3B, to show common enriched pathways downstream of Zinc and GR signaling.

We have performed gene list enrichment analysis on the overlap shown in figure 3B (48 genes) as requested by the reviewer. Significant enrichments (5% level, after multiple testing correction) are obtained using gene-function lists from bioplanet, wikipathways and KEGG, but not gene ontology (GO) terms. We found enrichments for Type II interferon signaling (interferon-gamma) [bioplanet & wikipathways], Interferon alpha/beta signaling [bioplanet],

*Interferon-gamma signaling pathway [bioplanet] and toxoplasmosis [KEGG]. Enrichments were done using the Enrich web-tool. The results of this relevant information have been added to **the results section** of the revised paper.*

5/ In figure 4A, bigger magnification images are needed in H&E staining. Furthermore, in mice treated with ZnSO₄ and injected with TNF, apart from the degranulation of Paneth cells a more complete characterization of their phenotype is needed. The authors should assess the Paneth/Goblet cell number, the expression of antimicrobial peptides, the intestinal epithelial cell permeability, as well as the epithelial cell death (apoptosis and necroptosis).

We agree with the reviewer.

We have done the following to ameliorate this part of the paper.

- 1. We have thawed MMP7^{-/-} embryo's, transferred them to fosters and at the age of 8W took an ileum sample of these mice as well as MMP7^{+/+} mice, to stain for MMP7 with the antibody and found indeed no staining in the former mice and nice staining in the latter. This control **was added** to the figure as a control. These data are added in the revised paper in the results section.*
- 2. We have increased the size of the H&E pictures and we also performed new experiments to study the effects of PBS or TNF injection in mice that had been treated for a week with H₂O or ZnSO₄ and made tissue sections for Transmission Electron Microscopy (TEM). The pictures are very revealing. They show very nicely the devastating cell-death effects of TNF on Paneth cells (extrusion in the lumen and loss of contact with the basal membrane) and that Zn is able to protect. These data support the cell death induction by TNF and the impact of Zn. They are **added in the revised paper** in the results section.*
- 3. We have also performed studies with mice treated with H₂O or ZnSO₄ for 1 week, after which mice were injected with either TNF or PBS (10 mice per group, total numbers n=40) and taken studied intestinal permeability by FITC dextran and taken ileum samples and done qPCR to detect Paneth cell specific markers qPCR, the decline of which being a readout of less PCs, as recently shown in the literature. These data confirm that PC numbers decline by TNF, but significantly less by Zn treatment. These **data are added** in the revised paper in the results section.*

6/ The resistance of GRDim, GRVillKO and Adx mice in ZnSO₄ treatment is attributed to the threshold of ISRE/IRF dependent gene expression. It might be worthwhile increasing the ZnSO₄ dosage in these mice to exclude a potential protective effect in higher doses.

This is an outstanding question, that we have been asking ourselves in the past. We have tries to address them for the revision of the paper by studying protective effects of higher doses of ZnSO₄ in ADX mice and in GRdim/dim mice. The C57BL/6J ADX mice, control C57BL/6J mice, as well as GRdim/dim and GRwt/wt mice were treated with 25 mM ZnSO₄ (the normal dose), 50 mM or 75 mM for one week, and were then challenged with a lethal dose of TNF. The ADX

*experiment failed since 50 mM ZnSO₄ as such already killed more than half of the available mice while 75mM ZnSO₄ appeared to be lethal for all ADX mice. The few mice that remained alive and were challenged with TNF showed no protection whatsoever. The GRdim/dim and GRwt/wt mice experiment was more informative. These mice all survived the three ZnSO₄ doses, although 50 and 75 mM caused visible distress to the mice (piloerection, less activity, diarrhea). In GRwt/wt mice, TNF toxicity was rescued by Zn by 25 mM, but less by the higher doses, probably because of toxic effects. In GRdim/dim mice we found (deaths on total) in the 0 mM, 25mM, 50 mM and 75 mM were 3/5, 3/4, 3/4 and 4/5. We have chosen not to show the data, but **to mention them in the paper.***

7/ The authors should revise their reference list including publications showing that Zinc can modulates the gut microbiota composition. (Zackular et al, Nature Medicine, 2016 & Dalin Li et al, Gastroenterology 2016).

We agree and added these references.

8/ Although there is a lot of literature on the anti-inflammatory role of Zinc, the authors report an independent function during TNF-induced SIRS. The data presented in SI Appendix, Figure S3, show that ZnSO₄ treatment although doesn't affect proinflammatory cytokines levels in the ileum, reduces IL-6 and Eotaxin serum levels. These results should be discussed further as IL-6 and Eotaxin are the major cytokines driving mortality in this model. In the same context, serum IL1 levels should also be measured.

*We have now also measured IL1b serum levels, and although they slightly increase after TNF challenge, they are not reduced by ZnSO₄ pretreatment. We have **added the data** to the paper and expanded our discussion.*

9/ In figure 5A statistical significance in the microbiota composition between wild type and ZNSO₄ treated mice should be indicated.

The statistical significance of the differences has been added to the figure.

10/ The working hypothesis as presented in figure 3I seems to be redundant as it is summarized in the graphical proposed mechanism (figure 5E).

Indeed, this has been corrected.

9th Jul 2020

Dear Claude,

Thank you for your patience and for the submission of your revised manuscript to EMBO Molecular Medicine. We have now received the enclosed reports from the referees that were asked to re-assess it.

As you will see, while referee 1 is now supportive, referee 2 still has some concerns that must be addressed in writing and the study limitation must be made clear in the discussion section of the article as indicated by referee 1 during our cross-commenting exercise: "I do agree that the lack of the specificity control (the transfer of a microbial strain, not depleted by ZnSO₄ that should not suffice to reverse the protective effect of ZnSO₄) would substantiate the claims made by the authors. Therefore, the authors should write three sentences in the discussion to state that they did not perform this important experiment and what this control would explain."

We will be able to accept your manuscript pending the above and following final amendments:

1) Please provide a point-by-point letter INCLUDING my comments as well as the reviewer's reports and your detailed responses to their comments (as Word file).

Please submit your revised manuscript within two weeks (if possible) or as soon as you will be back in the office.

Kind regards,

Celine

Celine Carret, PhD

Senior Editor

EMBO Molecular Medicine

***** Reviewer's comments *****

Referee #1 (Comments on Novelty/Model System for Author):

The authors used germ-free mouse models to demonstrate a protective role of the gut microbiota in the TNF-induced severe inflammatory response syndrome. They further show that the glucocorticoid receptor is critically involved in this protective role of zinc. This is an important and medically relevant message.

Referee #1 (Remarks for Author):

My previous concerns have been adequately addressed by the authors in their revised manuscript version. This revealed protective role of the microbiota in TNF-induced SIRS is interesting and this finding certainly fits with the scope of EMBO Molecular Medicine.

Referee #2 (Remarks for Author):

The revised manuscript has been improved. New RNA-seq data, transfer microbiota experiment and analysis of new genetically deficient mice have been added to the evidence in support of the proposed hypothesis. The additive protective effect of ZnSO₄ and dexamethasone (DEX), which is the most clinically significant result, is nicely represented in the in vivo TNF induced SIRS model and further supported by the new data added in Figure 7.

However, there are still mechanistically important issues that remain critical, yet unresolved. Specifically, the authors spent efforts to support a causal effect of the ZnSO₄ modified gut microbiota in the protection of mice from TNF induced lethality. However, the experimental setup described in the revised manuscript is still insufficient to provide a direct link. The microbiota transfer experiment in Figure 6 that involves transfer of *S. sciuri* and *S. nepalensis* (colonies depleted by ZnSO₄) in ZnSO₄ treated mice, re-sensitized them to TNF induced lethal shock. The experiment however, lacks a specificity control i.e the transfer of a microbial strain, not depleted by ZnSO₄ that should not suffice to reverse the protective effect of ZnSO₄.

In addition, the effect of ZnSO₄ treatment in restoring the TNF-induced dysfunction of Paneth cells and the epithelial cell death, presented on Figure 5B and C, would be more convincing if quantification could be performed (e.g. by cell death detection assays). Notably, intestinal permeability seems not to be affected by Zinc administration (Appendix, Figure S6), which is contradictory to the fact that Zinc reduces bacterial dissemination to the peripheral organs. This generates additional questions on the protective mechanisms of Zinc, under investigation in this work.

Collectively, although, I strongly believe that the manuscript addresses a very intriguing aspect of ZnSO₄ and DEX supplementation in diseases related to TNF-induced cell death, the revised manuscript still lacks direct experimental evidence in support of a coherent mechanism.

point-by-point letter

As you will see, while referee 1 is now supportive, referee 2 still has some concerns that must be addressed in writing and the study limitation must be made clear in the discussion section of the article as indicated by referee 1 during our cross-commenting exercise: "I do agree that the lack of the specificity control (the transfer of a microbial strain, not depleted by ZnSO₄ that should not suffice to reverse the protective effect of ZnSO₄) would substantiate the claims made by the authors. Therefore, the authors should write three sentences in the discussion to state that they did not perform this important experiment and what this control would explain."

Agreed. I have added this information in the discussion of the paper as: "More details on how *S. sciuri* and *S. nepalensis* contribute to ISRE/IRF upregulation will have to be unfolded by later studies. Also, monoculture transplantation studies using other bacterial strains, which were not depleted by ZnSO₄, or which will be identified by microbiota sequencing studies rather than by culturing, but will increase the specificity of our findings and make them more complete, in the future."

1) Please provide a point-by-point letter INCLUDING my comments as well as the reviewer's reports and your detailed responses to their comments (as Word file).

Please see this letter.

******* Reviewer's comments *******

Referee #1 (Comments on Novelty/Model System for Author):

The authors used germ-free mouse models to demonstrate a protective role of the gut microbiota in the TNF-induced severe inflammatory response syndrome. They further show that the glucocorticoid receptor is critically involved in this protective role of zinc. This is an important and medically relevant message.

Referee #1 (Remarks for Author):

My previous concerns have been adequately addressed by the authors in their revised manuscript version. This revealed protective role of the microbiota in TNF-induced SIRS is interesting and this finding certainly fits with the scope of EMBO Molecular Medicine.

Thank you.

Referee #2 (Remarks for Author):

The revised manuscript has been improved. New RNA-seq data, transfer microbiota experiment and analysis of new genetically deficient mice have been added to the evidence in support of the proposed hypothesis. The additive protective effect of ZnSO₄ and dexamethasone (DEX), which is the most clinically significant result, is nicely represented in the in vivo TNF induced SIRS model and further supported by the new data added in Figure 7.

However, there are still mechanistically important issues that remain critical, yet unresolved. Specifically, the authors spent efforts to support a causal effect of the ZnSO₄ modified gut microbiota in the protection of mice from TNF induced lethality. However, the experimental setup described in the revised manuscript is still insufficient to provide a direct link. The microbiota transfer experiment in Figure 6 that involves transfer of *S. sciuri* and *S. nepalensis* (colonies depleted by ZnSO₄) in ZnSO₄ treated mice, re-sensitized them to TNF induced lethal shock. The experiment however, lacks a specificity control i.e the transfer of a microbial strain, not depleted by ZnSO₄ that should not suffice to reverse the protective effect of ZnSO₄.

We agree with this interpretation of the reviewer and are happy that the editor provides us the chance to reply on this comment in the paper. Indeed, the finding that Zn modifies the microbiome, particularly the two Staphylococci species, point to part of the mechanism underlying its protective effects, but the entire picture will have to follow from additional, extensive studies, involving very deep NGS experiments of the ileum microbiota. These studies will be done over the next years. We have, as suggested by the editor, added comments on this in the Discussion section.

In addition, the effect of ZnSO₄ treatment in restoring the TNF-induced dysfunction of Paneth cells and the epithelial cell death, presented on Figure 5B and C, would be more convincing if quantification could be performed (e.g. by cell death detections assays). Notably, intestinal permeability seems not to be affected by Zinc administration (Appendix, Figure S6), which is contradictory to the fact that Zinc reduces bacterial dissemination to the peripheral organs. This generates additional questions on the protective mechanisms of Zinc, under investigation in this work.

I agree that there is a certain degree of contradiction (as I wrote in the previous point-to-point response). The FITC-dextran-assessed permeability increases by TNF but is not reduced by zinc, while the permeability assessed by bacterial transport from gut to periphery increases by TNF but is reduced by zinc, and the Paneth cell death is also clearly reduced by zinc. This issue needs further clarification in later work. We have now added the following to attract the attention of the reader to this point in the relevant section in the paper:

“Although the contribution of TNF-induced Paneth cell death in the degree of FITC-dextran-measured permeability assay is not known, the reduction in amount of colonizing bacterial species appears not to be directly determined by the general permeability of the gut, but by the composition of the ileum flora”.

Collectively, although, I strongly believe that the manuscript addresses a very intriguing aspect of ZnSO₄ and DEX supplementation in diseases related to TNF-induced cell death, the revised manuscript still lacks direct experimental evidence in support of a coherent mechanism.

We thank the reviewer for the support.

19th Aug 2020

Dear Claude

Thank you for resubmitting the manuscript so fast. We are pleased to inform you that your manuscript is accepted for publication and is now being sent to our publisher to be included in the next available issue of EMBO Molecular Medicine.

Please read below for additional IMPORTANT information regarding your article, its publication and the production process.

Congratulations on your interesting work!

Celine

Celine Carret, PhD
Senior Editor
EMBO Molecular Medicine

Follow us on Twitter @EmboMolMed
Sign up for eTOCs at embopress.org/alertsfeeds

Corresponding Author Name: Claude Libert

Manuscript Number: EMM-2019-11917